# An in vitro approach reveals molecular mechanisms underlying endocrine disruptor-induced epimutagenesis

**Jake D Lehle, Yu-Huey Lin, Amanda Gomez, Laura Chavez, John R McCarrey***

Department of Neuroscience, Developmental and Regenerative Biology, The University of Texas at San Antonio, San Antonio, United States

**Abstract** Endocrine disrupting chemicals (EDCs) such as bisphenol S (BPS) are xenobiotic compounds that can disrupt endocrine signaling due to steric similarities to endogenous hormones. EDCs have been shown to induce disruptions in normal epigenetic programming (epimutations) and differentially expressed genes (DEGs) that predispose disease states. Most interestingly, the prevalence of epimutations following exposure to many EDCs persists over multiple generations. Many studies have described direct and prolonged effects of EDC exposure in animal models, but many questions remain about molecular mechanisms by which EDC-induced epimutations are introduced or subsequently propagated, whether there are cell type-specific susceptibilities to the same EDC, and whether this correlates with differential expression of relevant hormone receptors. We exposed cultured pluripotent (iPS), somatic (Sertoli and granulosa), and primordial germ cell-like (PGCLC) cells to BPS and found that differential incidences of BPS-induced epimutations and DEGs correlated with differential expression of relevant hormone receptors inducing epimutations near relevant hormone response elements in somatic and pluripotent, but not germ cell types. Most interestingly, we found that when iPS cells were exposed to BPS and then induced to differentiate into PGCLCs, the prevalence of epimutations and DEGs was largely retained, however, >90% of the specific epimutations and DEGs were replaced by novel epimutations and DEGs. These results suggest a unique mechanism by which an EDC-induced epimutated state may be propagated transgenerationally.

***For correspondence:**
john.mccarrey@utsa.edu

**Competing interest:** The authors declare that no competing interests exist.

## eLife assessment

This **important** study, characterizing the epigenetic and transcriptomic response of a variety of cell types representative of somatic, germline, and pluripotent cells to BPS, reveals the cell type-specific changes in DNA methylation and the relationship with the genome sequence. The findings are **convincing** and provide a basis for future analyses in vivo. This work should be of interest to biomedical researchers who work on epigenetic reprogramming and epigenetic inheritance.

## Introduction

It has now been more than a half-century since Roy Hertz in 1958 first proposed the notion that certain chemicals, particularly those used in livestock feed at the time, could contaminate food sources and bioaccumulate in humans mimicking the activity of hormones (*Gassner et al., 1958*). As the number of potential chemicals that could exert these effects has grown, so too has the interest in this area to the point that this is now a major topic of active research. The dangers of many endocrine disrupting chemicals (EDCs) studied to date have largely been investigated utilizing animal models. A primary example is the multi-phased Toxicant Exposures and Responses by Genomic and Epigenomic

Regulators of Transcription (TaRGET) program established to determine the contribution of environmental exposures to disease pathogenesis as a function of epigenome perturbation in mouse models (*Wang et al., 2018*). While the use of animal models has been quite informative for determining the various systemic or organ/tissue-specific disease-related effects associated with exposure to different EDCs, the molecular mechanisms underlying these phenomena remain poorly understood. Thus, questions persist regarding (i) the mechanisms by which exposure of cells to EDCs results in the initial formation of epimutations defined here as any change in epigenomic programming, (ii) potential cell-type specific differential susceptibility to epimutagenesis induced by different EDCs, (iii) the potential involvement of relevant hormone receptors and/or genomic hormone response elements in EDC-induced disruption of the epigenome, and (iv) the ability of an EDC-induced epimutated state to persist inter- or transgenerationally despite generational epigenetic reprogramming (*Diaz-Castillo et al., 2019*).

Questions about molecular mechanisms responsible for EDC-exposure based disruption of normal epigenetic programming have been difficult to interpret in whole animal models due, in part, to the inherent complexity of the intact animal including potential interactions among multiple different organs, tissues, and cell types at the paracrine, metabolic and/or systemic levels. To circumvent this challenge, we opted to expose homogeneous populations of specific cell types in culture to doses of the EDC, Bisphenol S (BPS), that are below the maximum safe limit previously set by the EPA for exposure of humans to the similar estrogen mimetic, Bisphenol A (BPA). In this way, we hoped to learn more about the manner in which direct exposure to a specific EDC induces epimutations and differential gene expression in different individual cell types including somatic cells, pluripotent cells, and germ cells. This approach was designed to distinguish the direct effects of an EDC exposure in vitro from indirect effects that may accrue as the result of an EDC exposure on one cell type inducing a secondary effect on a neighboring or related cell type in vivo.

Previous in vitro studies have demonstrated direct susceptibility of cultured cell types to EDC-induced epimutagenesis, however, those studies were focused on immortalized cancer cell lines that do not necessarily model key cell types involved in normal initial incursion or subsequent intragenerational propagation or inter- or transgenerational transmission of environmentally induced epimutations in vivo (*Deb et al., 2016*; *Goodman et al., 2014*; *Huang et al., 2019*; *Karaman and Ozden, 2019*; *Senyildiz et al., 2017*). We chose to examine direct exposure of three key types of cells relevant to environmental exposures and subsequent propagation and transmission of induced epimutations in vivo – somatic cell types known to be responsive to endocrine signaling (Sertoli cells and granulosa cells), pluripotent cells mimicking the preimplantation embryo (iPSCs), and germline cells (PGCLCs).

While there have been reports that have clearly indicated the ability of EDCs to disrupt classical endocrine signaling (*Henley and Korach, 2010*; *Kelce et al., 1995*; *Swedenborg et al., 2009*; *vom Saal and Hughes, 2005*; *You et al., 1998*), there exist other reports indicating EDCs can also disrupt non-classical endocrine signaling via nuclear receptors (*Ozgyin et al., 2015*; *Myers et al., 2011*), G protein-coupled receptors (*Thomas and Dong, 2006*), and calcium channel signaling (*Brenker et al., 2018*). This has further complicated our understanding of the mechanism(s) by which EDC exposure initially induces epimutations, and has particularly confounded insight into the susceptibility of germ cells to EDC-induced epimutagenesis given that germ cells are reported to lack expression of classical endocrine receptors (*Meccariello et al., 2014*). This is relevant to the effort to understand mechanisms underlying transgenerational epigenetic inheritance of EDC-induced epimutations (*Anway et al., 2006*; *Guerrero-Bosagna et al., 2012*; *Nilsson et al., 2008*) which clearly implicates transmission via the germ line.

The concept of germline transmission of EDC-induced epimutations is further confounded by the known epigenetic reprogramming events that occur in the preimplantation embryo and developing fetal and neonatal germ line. Thus, regardless of how EDC-induced epimutations become initially manifest in the germ line, it is not clear how they subsequently persist and are transmitted inter- or transgenerationally given the large portion of the epigenome that undergoes genome-wide erasure and resetting of epigenetic programming during each generation (*Cantone and Fisher, 2013*; *Lee et al., 2014*; *Santos et al., 2002*; *Sanz et al., 2010*). In vitro cell culture systems afford the opportunity to focus on individual cell types independently, including somatic cells which are likely initially exposed to most environmental disruptive influences, pluripotent cells which represent the preimplantation embryo in which the first major reprogramming event occurs during normal development,

and early germline cells where the second major reprogramming event is initiated in primordial germ cells (PGCs). Importantly, in addition to the potential to study each of these cell types independently, it is possible to induce transitions in cell fate in vitro that model those that occur during normal development, thereby recapitulating normal reprogramming events in a way that can facilitate high resolution studies of the fate of EDC-induced epimutations once they have been induced in any of these cell types.

Here we describe our study of the relative susceptibility to induction of epimutations by direct exposure of four different cell types maintained in culture – somatic Sertoli and granulosa cells, pluripotent iPSCs, and germline PGCLCs – to BPS, followed by our analysis of the fate of epimutations initially induced in mouse iPSCs that are then induced to transition into PGCLCs. We found that there are cell-type specific differences in susceptibility to epimutagenesis and associated dysregulation of gene expression patterns following exposure of these different cell types to a similar dose of BPS below the maximum safe limit established by the EPA for exposure of human cells to BPA (*Ribeiro et al., 2019*). We further found that BPS induction of epimutations in both pluripotent and somatic cell types that express relevant estrogen receptors (ERs), as well as in germ cells which do not express ERs, suggests disruption of both canonical and non-canonical endocrine signaling. Most interestingly, we found that when iPSCs exposed to BPS were then induced to undergo a major transition in cell fate and related epigenetic reprogramming to form PGCLCs, a similar prevalence of epimutations and

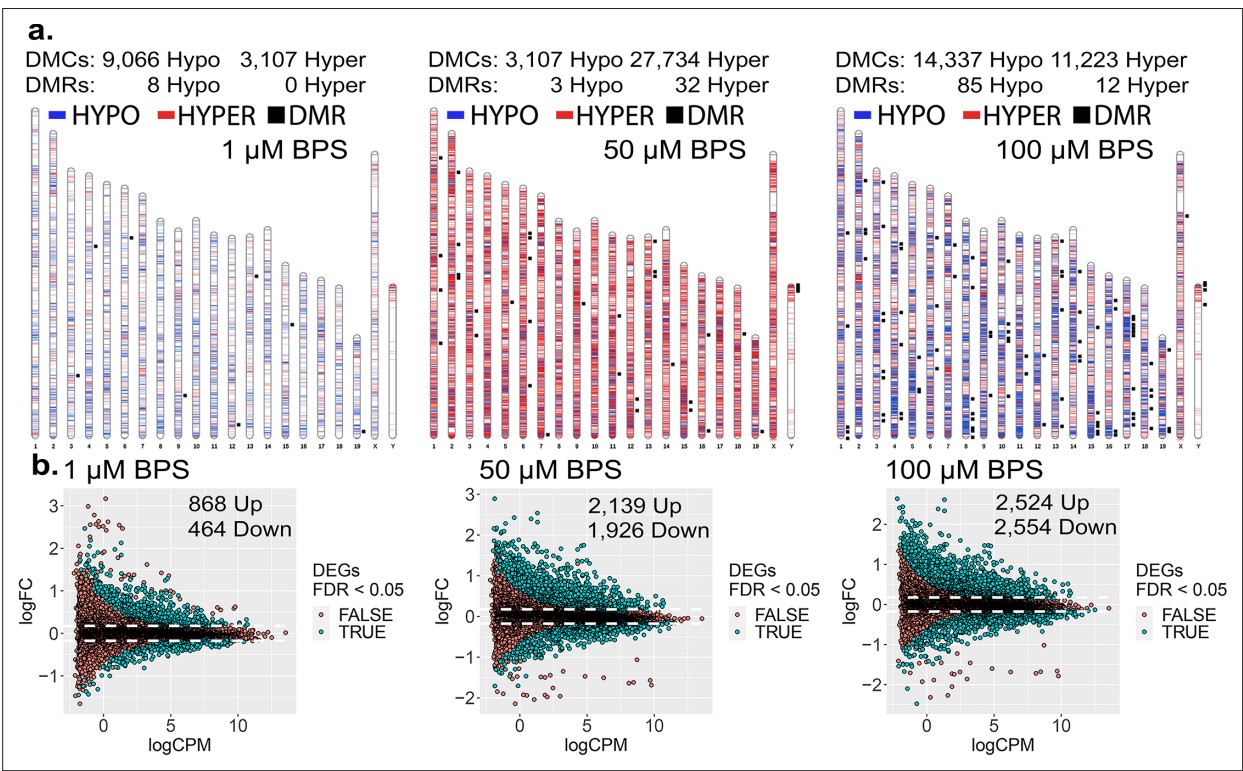

**Figure 1.** Dose-dependent impact of epimutagenesis measured in iPSCs exposed to 1, 50, and 100 µM BPS. (**a**) Ideogram plots displaying chromosomal distribution of genome-wide changes in DNA methylation caused by BPS exposure. (**b**) Mean difference (MD) plots of changes in gene expression following exposure to increasing doses of BPS. Exposure to increasing doses of BPS induced higher, although plateauing numbers of DMCs, DMRs, and DEGs. Blue horizontal lines = hypomethylated DMCs, red horizontal lines = hypermethylated DMCs, black squares = DMRs.

The online version of this article includes the following figure supplement(s) for figure 1:

**Figure supplement 1.** Overlapping DMCs and DEGs found among dose-dependent responses to BPS exposure.

**Figure supplement 2.** Relationship between DMCs at promoters and DEGs.

**Figure supplement 3.** Chemical exposure experimental workflow.

**Figure supplement 4.** Consistency among iPSC replicates and variation between RNA-seq and DNA methylation Infinium Beadchip array experimental and control groups.

**Figure supplement 5.** ICC validation of MF5-9-1 iPSCs.

**Table 1.** DEGs containing DMCs observed in iPSC exposed to increasing doses of BPS.

| DEGs containing DMCs | iPSC 1 µM | iPSC 50 µM | iPSC 100 µM |
|---|---|---|---|
| Promoter | 264 (19.82%) | 693 (17.04%) | 1136 (22.37%) |
| Gene body | 436 (32.73%) | 1541 (37.91%) | 1934 (38.08%) |

DEGs was retained, but, surprisingly, very few (<10%) specific epimutations or DEGs were conserved during this process. This suggests that the initial EDC exposure induces disruption of the chromatin landscape that subsequent epigenetic reprogramming is unable to fully restore. Thus, reprogramming during a major cell fate transition appears to correct many of the initially induced epimutations, but also appears to induce many de novo epimutations, and this imbalance may persist across multiple generations which may contribute to continued transgenerational epigenetic inheritance of EDC-induced phenotypes during succeeding generations.

## Results

### Dose-dependent epimutagenesis response to BPS exposure in iPSCs

To initially assess the extent to which epimutations and dysregulated gene expression can be induced in cells maintained in culture, we exposed mouse iPSCs to three doses of BPS (1, 50, and 100 µM) and measured the impact on the epigenome and transcriptome using the Illumina Infinium Mouse Methylation BeadChip Array and RNA-seq, respectively (*Figure 1*). All three doses of BPS induced individual differentially methylated CpGs (DMCs) as well as differentially methylated regions (DMRs) (*Figure 1a*), and differentially expressed genes (DEGs) (*Figure 1b*) when compared to control mouse iPSCs treated with vehicle (EtOH) only. The extent of this exposure-specific epimutagenesis was correlated with the dose of BPS used. Information regarding the overlap of DMCs/DMRs/DEGs identified for each dose of BPS is shown in *Figure 1—figure supplement 1*. We observed BPS-induced DMCs within both promoter and gene body regions of a portion of DEGs (17-38%) (*Table 1*). Importantly, in many cases, we observed a correlation between differential expression of a gene and the presence of DMCs in the promoter region of that gene (*Figure 1—figure supplement 2a*). These results provided an initial proof of concept that exposure of cells maintained in vitro to an EDC such as BPS can disrupt the epigenome and transcriptome in a dose-dependent manner. Interestingly, exposure to 1 µM BPS induced very few DMRs, but did induce widespread DMCs and DEGs. Because 1 µM BPS is below the FDA's suggested safe environmental level established for exposure of humans or intact animals to BPA (*Ribeiro et al., 2019*), and was sufficient to induce DMCs and dysregulated gene expression on all chromosomes in our cultured iPSCs, we utilized this dose and focused solely on DMCs when assessing epimutations in all subsequent experiments (*Table 1*).

### Cell-type specific susceptibility to induction of epimutations following BPS exposure

We next sought to determine if different key cell types – somatic, pluripotent or germ – are differentially susceptible to induction of epimutations in response to direct exposure of each to a similar dose of BPS. Thus, we exposed pluripotent (iPSCs), somatic (Sertoli and granulosa cells), and germ (PGCLCs) cell types to 1 µM BPS and measured changes in DNA methylation patterns. We identified exposure-specific DMCs in each exposed cell type relative to its corresponding control (exposed to carrier only; *Figure 2a*). We observed overall differences among the different cell types in total numbers of DMCs, with iPSCs showing the highest number of DMCs, followed by Sertoli cells and granulosa cells, and then PGCLCs, respectively (*Table 2*).

Interestingly, among the exposure-specific DMCs identified in each cell type, we observed predominantly hypomethylated DMCs in the somatic and pluripotent cell types, but predominantly hypermethylated DMCs in PGCLCs (*Table 2*). These findings confirm that there are both quantitative differences among cell types in overall susceptibility to epimutagenesis, and qualitative cell-type specific differences in the prevalence of hypo- versus hypermethylated DMCs following exposure to BPS. The latter likely reflects the fact that the epigenome in PGCs is naturally more hypomethylated

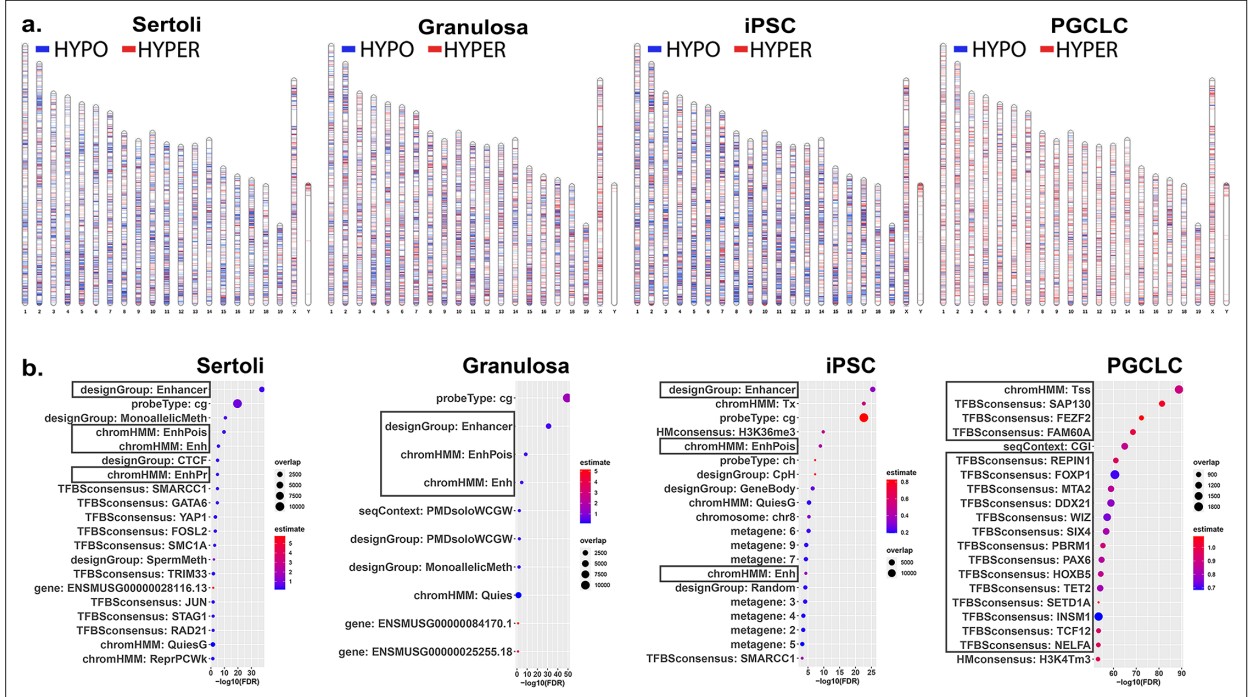

**Figure 2.** Chromosomal distributions and annotations of BPS-induced epimutations in pluripotent, somatic, and germ cell types. (**a**) Ideograms illustrating chromosomal locations of DMCs induced by exposure of each cell type to 1 μM BPS. Blue horizontal lines = hypomethylated DMCs, red horizontal lines = hypermethylated DMCs. (**b**) Enrichment plots indicating feature annotations in genomic regions displaying prevalent BPS-induced epimutations in each cell type. Dot size = number of overlapping DMCs with specific annotation, dot color = enrichment score reflecting the relative degree to which epimutations occurring in a specific annotated class are overrepresented.

The online version of this article includes the following figure supplement(s) for figure 2:

**Figure supplement 1.** Overlapping DMCs and DEGs found among cell-type specific responses to BPS exposure.

**Figure supplement 2.** ICC control staining of cell type-specific markers.

**Figure supplement 3.** FACS sorting for ITGB3/FUT4 enriched primordial germ-cell like cells.

**Figure supplement 4.** Quality control for Infinium Mouse Methylation BeadChip Array data.

than that in pluripotent or somatic cell types (*Hajkova, 2011*), enhancing the likelihood that most changes in DNA methylation induced in PGCLCs will necessarily involve hypermethylation.

## Annotation of BPS-induced epimutations

As we only found an average of 11.05% direct overlap in DMCs between two or more cell types and no overlapping DMCs shared between all cell types (*Figure 2—figure supplement 1*), we next analyzed annotations associated with genomic sites of BPS exposure-specific DMCs in each cell type. We mined annotations associated with genomic regions included on the Infinium array and found that a substantial group of exposure-specific DMCs was associated with enhancer regions in the two somatic (Sertoli and granulosa) and one pluripotent (iPS) cell types exposed to BPS, while in PGCLCs

**Table 2.** Treatment-specific differentially methylated sites (DMCs) (treated vs. control).

| DMCs | Sertoli | Granulosa | iPSCs | PGCLCs |
|---|---|---|---|---|
| Hypomethylated* | 7385 | 6444 | 9651 | 2315 |
| Hypermethylated† | 3022 | 4143 | 4308 | 4785 |
| Total | 10,407 | 10,587 | 13,959 | 7100 |

* A CpG site that was predominantly methylated in the control samples but unmethylated in the exposed samples.
†A CpG site that was predominantly unmethylated in the control samples but methylated in the exposed samples.

BPS-induced DMCs were more prevalent at promoter regions containing transcription start sites (TSSs) and transcription factor binding sites (*Figure 2b*), indicative of yet another qualitative cell-type specific difference in induction of epimutations by the same dose of BPS.

## Relationship between susceptibility to BPS-induced epimutagenesis and expression of relevant hormone receptors

As EDCs are thought to induce epimutations via disruption of classical hormonal signaling (*Henley and Korach, 2010*; *Kelce et al., 1995*; *Swedenborg et al., 2009*; *vom Saal and Hughes, 2005*; *You et al., 1998*), we next sought to determine if the differential extent of BPS induction of epimutations we observed in different cell types was associated with differential expression of relevant hormone receptors in each. We performed immunocytochemistry (ICC) to detect the presence of the relevant estrogen receptors – ERα and ERβ, while co-staining for cell-type specific markers (WT1 [Sertoli cell marker], FSHR [granulosa cell marker], FUT4 [iPSC marker], and NANOG [PGCLC or endogenous PGC marker]) to confirm the identity of the four cultured cell types examined in this study (Sertoli cells, granulosa cells, iPSCs and PGCLCs), as well as endogenous mouse PGCs (*Figure 3a*). Both somatic cell types (Sertoli and granulosa) showed positive immunolabeling for both ERα and ERβ, whereas the pluripotent cells showed positive immunolabeling for ERβ, but negative immunolabeling for ERα, and the PGCLCs and endogenous PGCs showed negative immunolabeling for both ERα and ERβ. Importantly, the latter result is consistent with previous reports of lack of expression of either ERα or ERβ at the protein level in endogenous PGCs (*Meccariello et al., 2014*).

These results demonstrate cell-type specific differences in expression of hormone receptors (ERα and ERβ) which are potentially relevant to the disruptive action of the estrogen mimetic, BPS. We noted that expression of at least one potentially relevant hormone receptor (ERβ) in either somatic or pluripotent cells correlated with a higher incidence of DMCs induced by exposure of somatic or pluripotent cell types to 1 μM BPS relative to the incidence of DMCs induced by exposure of PGCLCs, which do not express either estrogen receptor, to the same dose of BPS (*Table 2*). Nevertheless, we did still observe induction of DMCs when PGCLCs were exposed to BPS, demonstrating that expression of relevant canonical hormone receptors is not an absolute requirement for induction of epimutations in response to an EDC. To exclude the possibility that the observed susceptibility in germ cells could be correlated with the presence of other endocrine receptors that could interact with BPS, we performed additional ICC for the presence of AR, PPARγ, and RXRα and found they were all absent as well in endogenous PGCs (*Figure 3—figure supplement 1*). This result and our observation noted above that somatic and pluripotent cell types showed a higher incidence of epimutations at apparent enhancer regions while germ cells showed a higher epimutation incidence at apparent promoter regions reveal cell-type specific differences in susceptibility to induction of epimutations by exposure to the EDC, BPS.

## Proximity of BPS-induced epimutations to genomic EREs

The relatively higher incidence of BPS-induced epimutations in cell types expressing one or both estrogen receptor(s) suggests BPS-induced epimutagenesis may be manifest, at least in part, through canonical endocrine signaling pathways. If this is the case, we might expect to see elevated induction of epimutations in genomic regions enriched for relevant HREs which, for interactions with the estrogen mimetic, BPS, would be EREs. Previous studies have defined a full ERE consensus sequence (*Bourdeau et al., 2004*), but other reports have indicated that estrogen receptors can often bind to ERE half-sites (*Mason et al., 2010*; *Figure 3b*). Indeed, when we mined publicly available ERα ChIP-seq peaks from the UCSC genome browser database (*Dunham and Kundaje, 2012*; *Myers et al., 2011*; *Sloan et al., 2016*; *Wang et al., 2013*; *Wang et al., 2012*) and performed motif enrichment, we identified two distinct ERE half-sites within regions enriched for DMCs rather than one full-sized ERE consensus sequence, with the half-sites appearing to be more biologically relevant to interaction with estrogen or its mimetics (*Figure 3b*). We then plotted the frequency of ERE half-sites in genomic regions within 500 bp of BPS-induced epimutations (*Figure 3c and d*). We found an increase in the frequency of ERE half-sites identified within 500 bp of BPS-induced DMCs genome-wide in all four cell types investigated, but the frequency was notably lower in germ cells (*Figure 3c*). Thus, this higher frequency of ERE half-sites within 500 bp of BPS-induced DMCs was conserved in three of the four cell types – somatic (Sertoli and granulosa) and pluripotent (iPSCs) when we focused solely on apparent

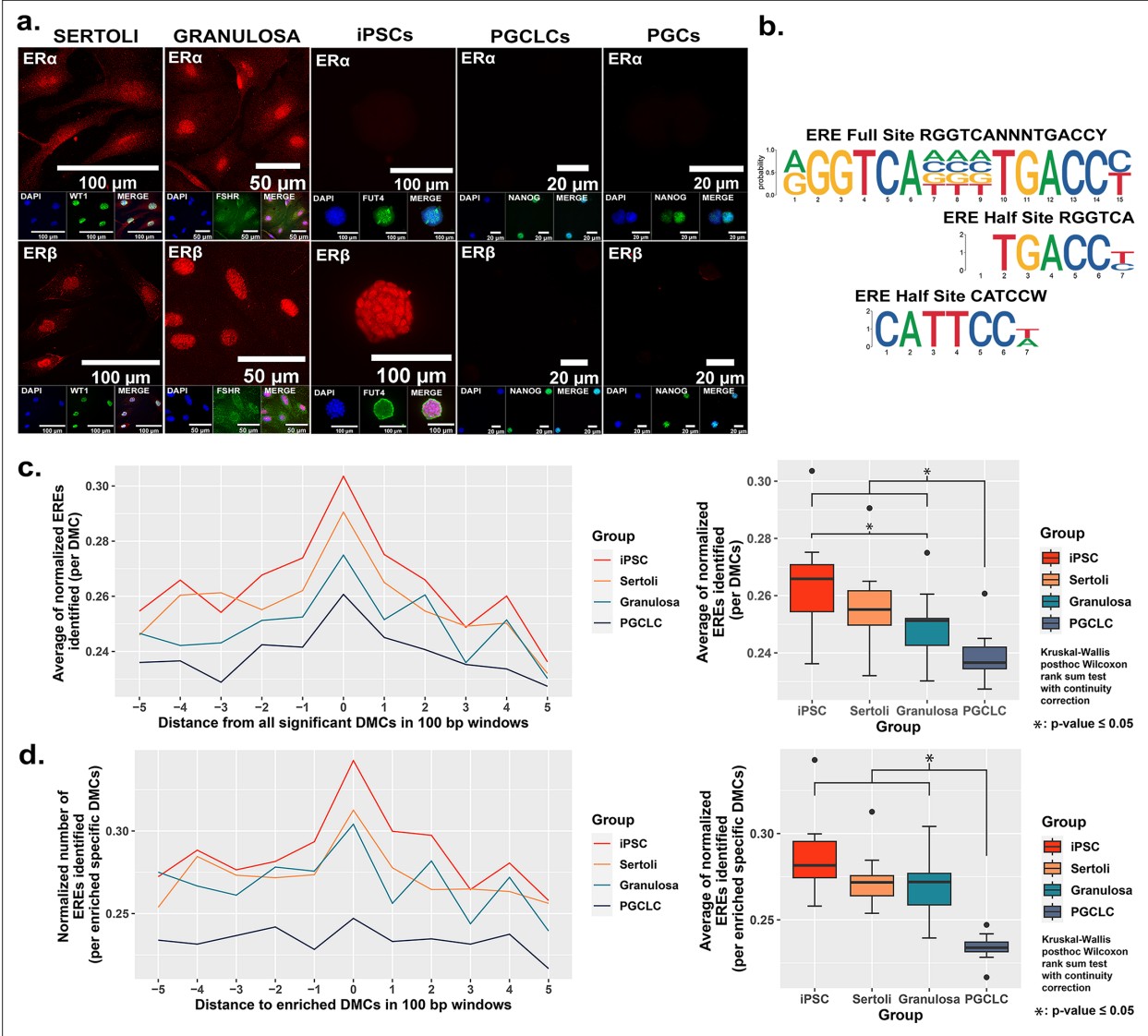

**Figure 3.** Correlation between cell-type specific expression of estrogen receptors and density of genomic EREs associated with BPS-induced epimutations. (**a**) Assessment of expression of ERα and ERβ by cell types co-stained for known cell-type specific markers. Somatic cell types express both receptors, pluripotent cells express ERβ but not ERα, and germ cells do not express either estrogen receptor. (**b**) Motif plots displaying the full ERE consensus sequence and the more biologically relevant ERE half-site motifs found to be enriched from ERα ChIP-seq. (**c,d**) Normalized density plots and box plots displaying the frequency of ERE half-sites identified (**c**) within 500 bp of all BPS-induced DMCs genome-wide, or (**d**) within 500 bp of the most enriched categories of BPS-induced DMCs in each cell type (=enhancer regions for somatic and pluripotent cell types and promoter regions in the germ cell type).

The online version of this article includes the following figure supplement(s) for figure 3:

**Figure supplement 1.** Assessment of expression of additional endocrine receptors potentially involved in cell type-specific responses to BPS exposure.

**Figure supplement 2.** Motifs near enriched DMCs.

**Figure supplement 3.** Comparison of delta beta values at significant DMCs.

**Figure supplement 4.** Genome-wide annotation of ERE half-sites.

enhancer regions, but not in germ cells (PGCLCs) where the majority of DMCs occurred at apparent promoter regions (*Figure 3d*). We also mined the UCSC mouse genome sequence for genome-wide prevalence of ERE half-sites and found that while these sites occur in both promoter and enhancer regions, they are nearly fourfold more frequent in enhancers (*Figure 3—figure supplement 4* and *Table 3*). Thus, it is perhaps not surprising that BPS-induced epimutations were more prevalent in

**Table 3.** Summary of ERE annotations.

| CpG islands | Repeat regions | Gene bodies | Promoters | Enhancers |
|---|---|---|---|---|
| 25,079 | 2,631,743 | 2,448,668 | 172,707 | 468,072 |

enhancer regions in somatic and pluripotent cell types that express one or both ERs, than in germ cells that do not express either ER.

Taken together, these results suggest a relationship between (i) expression of either ERβ alone or ERα and ERβ together, (ii) elevated susceptibility to BPS-induced epimutagenesis, and (iii) the occurrence of BPS-induced DMCs in genomic regions – particularly enhancers – containing EREs. These observations are consistent with the notion that one mechanism contributing to EDC-based induction of epimutations involves disruption of canonical endocrine signaling pathways. However, the fact that exposure to BPS also induced epimutations in germ cells, even in the absence of expression of estrogen-related receptors, and generated many DMCs in regions not inclusive of EREs, suggests that disruption of canonical endocrine signaling pathways is not the only mechanism by which exposure to an EDC can induce epimutations.

## Cell-type specific features of BPS-induced epimutations

To further interrogate cell-type specific differences in the genesis of epimutations following exposure to BPS, we compared DNA methylation patterns detected in the control (vehicle only) samples to identify naturally occurring, cell-type specific DMCs inherently associated with each different cell fate. Interestingly, of the 297,415 distinct CpGs interrogated by the Illumina Infinium Mouse Methylation BeadChip Array used for this analysis, >240,000 showed some degree of inherent differential methylation among the four cell types tested, suggesting they are tied to cell-fate specific differential DNA methylation (*Figure 4a*). We next determined the extent to which DMCs induced specifically by exposure of each cell type to BPS occurred at CpG dinucleotides that were among these naturally occurring cell-type specific DMCs. We found that a large majority of BPS exposure-specific epimutations detected in each cell type occurred at CpGs that also show inherent, cell-type specific variation in DNA methylation (>95% of BPS-induced epimutations in somatic and pluripotent cell types and ~89% in germ cells; *Figure 4a*). As with the overall pattern of BPS-induced epimutations shown in *Figure 2*, BPS-induced epimutations at CpGs showing inherent cell-type specific variation were enriched in apparent enhancer regions that occurred near EREs in somatic and pluripotent cell types, whereas those in germ cells were found predominantly in promoter regions containing significantly fewer EREs (*Figure 4b and d*). However, the low percentage of BPS-induced epimutations that occurred at CpGs that did not show inherent cell-type specific variation in DNA methylation patterns were found to be enriched in promoter regions lacking nearby EREs in all four cell types (*Figure 4c and d*). Thus, this latter group of BPS-induced epimutations appears to represent a small core group that arises following exposure to BPS via a mechanism that does not rely upon disruption of canonical endocrine signaling, and so is common to all cell types, regardless of expression of relevant endocrine receptors or nearby residence of relevant HREs within the genome.

## Impact of BPS exposure on gene expression

We next sought to determine if the cell-type specific differential susceptibility to BPS-induced epimutagenesis translated to a similar extent of dysregulation of gene expression in each cell type. We found that there was a similar relationship between the presence of DMCs in the promoter region of a gene and differential expression of that gene in all four cell types examined (*Figure 1—figure supplement 2b*). Somewhat surprisingly, RNA-seq analysis of gene expression patterns in each exposed cell type relative to its corresponding control (same cell type exposed to carrier only) revealed the greatest number of dysregulated genes in PGCLCs, despite PGCLCs being the cell type that showed the lowest number of epimutations following exposure to BPS (*Tables 2 and 4*). iPSCs showed the second highest level of dysregulated genes, while somatic Sertoli and granulosa cells showed relatively low levels of dysregulated gene expression (*Table 4*). Indeed, numbers of dysregulated genes were three orders of magnitude lower in differentiated somatic cells than in either pluripotent cells or germ cells (*Table 4*).

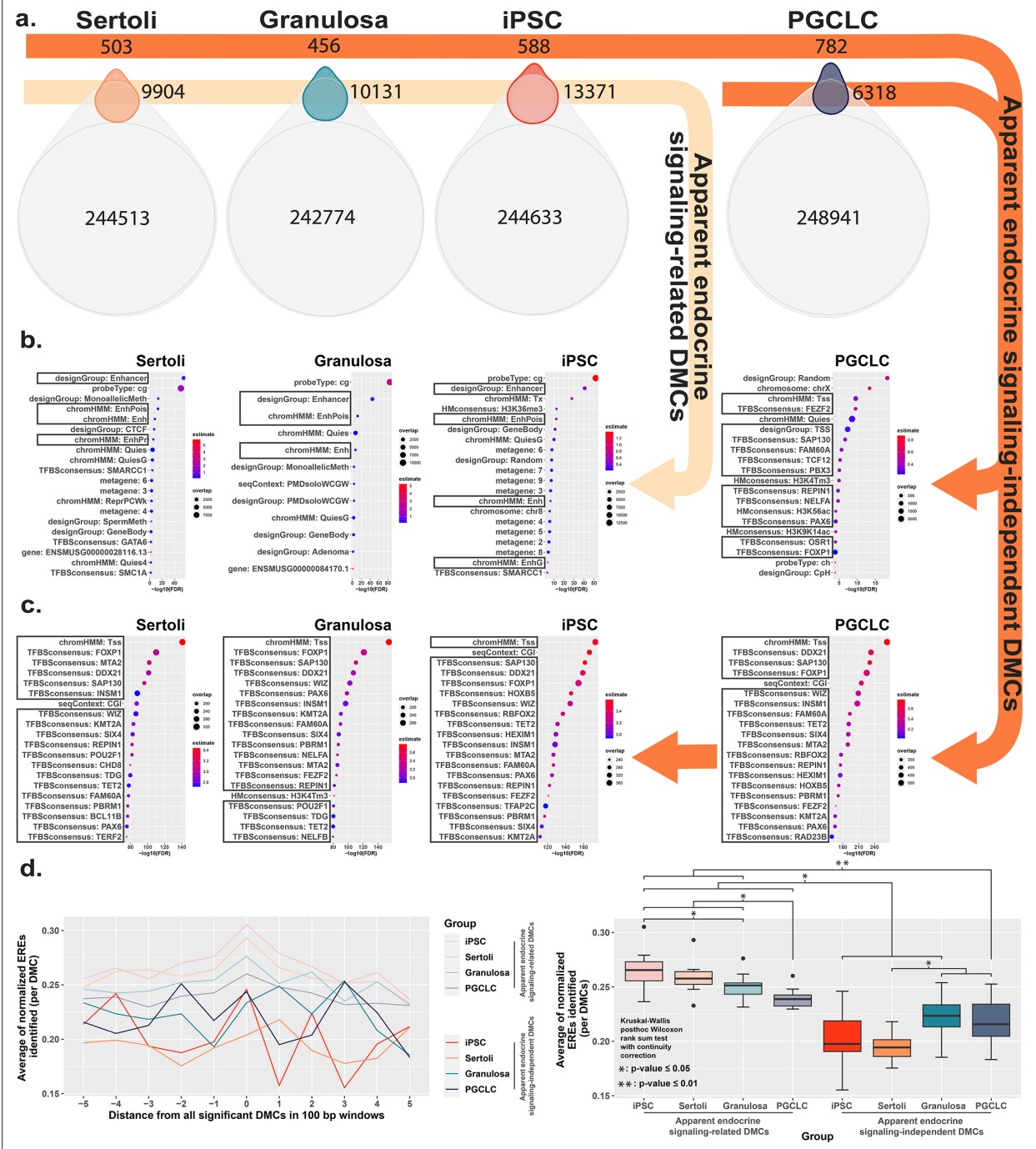

**Figure 4.** Direct comparison of BPS exposure-specific and cell-type specific features between cell types. (**a**) Assessment Venn diagrams indicating DMCs that are due either to BPS exposure (top, smaller ovals) or inherent cell-type specific differences (bottom, larger ovals). Numbers of apparent endocrine-signaling related DMCs are shown in the light orange arrow, and apparent endocrine-signaling independent DMCs are shown in the dark orange arrows. Enrichment plots indicating feature annotations in genomic regions displaying (**b**) apparent endocrine-signaling related DMCs occurring predominantly in enhancer regions in somatic Sertoli and granulosa cell types or pluripotent cells expressing one or more estrogen receptors, or (**c**) a smaller set of apparent endocrine-signaling independent DMCs occurring predominantly in promoter regions in all four cell types regardless of +/-expression of relevant endocrine receptors. (**d**) Normalized density plots and box plots displaying the frequency of ERE half-sites identified within 500 bp of apparent endocrine-signaling related DMCs occurring predominantly in enhancer regions and apparent endocrine-signaling independent DMCs occurring predominantly in promoters.

**Table 4.** Exposure-specific differentially expressed genes*.

| DEGs | Sertoli | Granulosa | iPSCs | PGCLCs |
|---|---|---|---|---|
| Down-regulated | 3 | 0 | 343 | 844 |
| Up-regulated | 32 | 2 | 694 | 1046 |
| Total | 35 | 2 | 1037 | 1890 |

*Genes showing significant differential expression following exposure of each cell type to 1 μM BPS relative to matched control cell types exposed to carrier only.

Because BPS-exposed PGCLCs showed the highest level of dysregulated genes, as well as the highest enrichment of DMCs occurring primarily at promoters (*Figure 2b*), we next assessed the general proximity between DMCs and promoter regions in each cell type to determine if DMCs in promoter regions may be more likely to predispose dysregulated gene expression than DMCs elsewhere in the genome (*Figure 5a*). Indeed, we found that in all four cell types, the smaller the median distance between BPS-induced DMCs and neighboring promoter regions, the larger the number of BPS-induced DEGs (*Figure 5b*). Thus, it appears that while PGCLCs showed the fewest overall BPS-induced DMCs among the four cell types exposed to BPS, this exposure induced a higher proportion of epimutations in regions within or adjacent to promoters in this cell type. This, and the relatively high extent of decondensed chromatin genome-wide in fetal germ cells, appear to have contributed to the higher incidence of dysregulated gene expression in BPS-exposed PGCLCs than that observed in the other three BPS-exposed cell types, even though the overall numbers of BPS-induced epimutations were greater in the other cell types.

## Potential non-canonical signaling pathways disrupted by BPS exposure

To identify potential mechanisms by which BPS exposure may induce epimutations via disruption of non-canonical endocrine signaling pathways, we mined our RNA-seq data to identify genes dysregulated independently of expression of relevant hormone receptors or presence of nearby EREs. We identified a set of genes in all four cell types tested that displayed promoters enriched for apparent endocrine-signaling independent DMCs (*Figure 6—figure supplement 1*). Gene ontology (GO) analysis indicated that several of the 1957 genes we identified were associated with ubiquitin-like protease

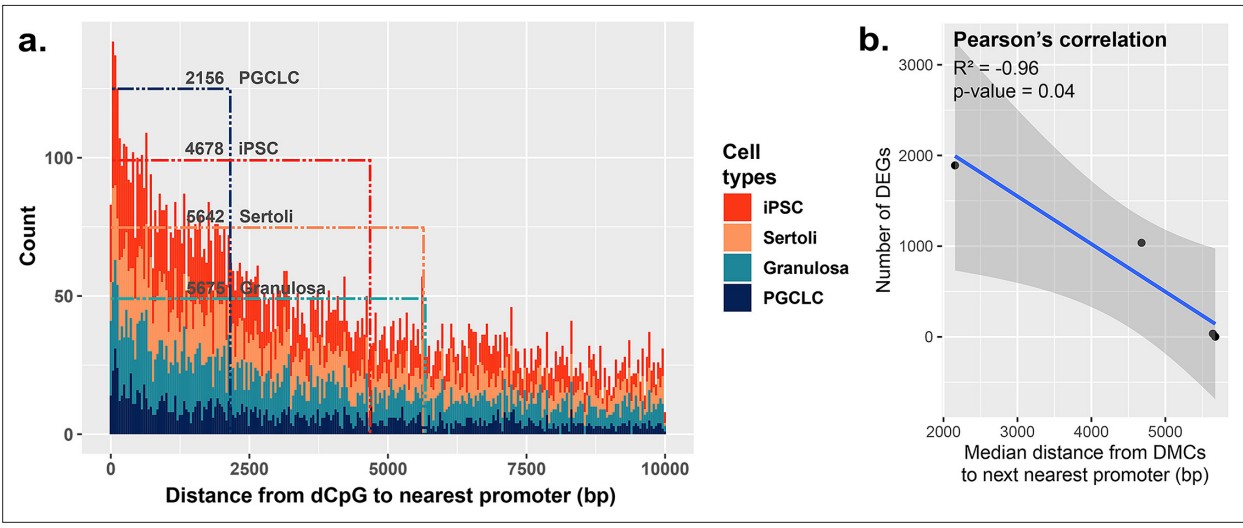

**Figure 5.** Correlation between the proximity of DMCs to promoters and dysregulation of gene expression. (**a**) Proximity plot displaying distances from exposure-specific DMCs to nearest promoter regions. Dotted lines indicate median points of the data for each cell type. (**b**) Correlation plot displaying a negative relationship between the distance from DMCs to nearest promoters and resulting dysregulation of gene expression within each cell type.

The online version of this article includes the following figure supplement(s) for figure 5:

**Figure supplement 1.** Consistency among replicates of pluripotent, somatic, and germ cell types and variation between DNA methylation Infinium Beadchip array experimental and control groups.

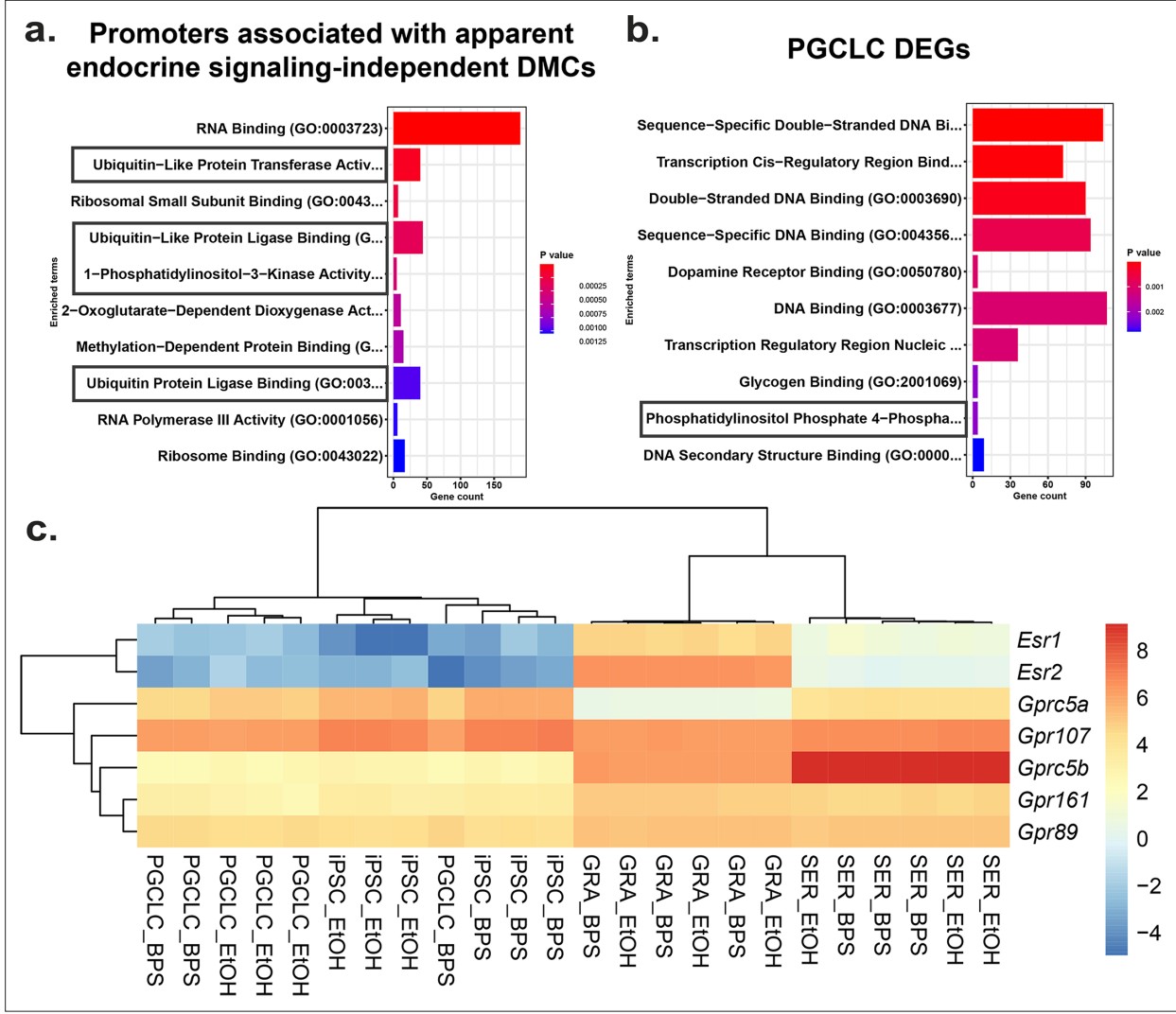

**Figure 6.** Potential involvement of non-canonical estrogen signaling pathways in BPS-induction of epimutations. Relative expression of genes (**a**) enriched for apparent endocrine-signaling independent promoter-region DMCs found in all cell types or (**b**) dysregulated in PGCLCs which lack expression of estrogen receptors. (**c**) Heatmap of relative expression of estrogen receptor genes (*Esr1* and *Esr2*) and G-coupled protein receptors (*Gprc5a*, *Gpr107*, *Gprc5b*, *Gpr161*, and *Gpr89*) in pluripotent, somatic, and germ cell types. *Gprc5a*, *Gpr107*, *Gprc5b*, *Gpr161*, and *Gpr89* all have been shown to bind to BPA or 17β-estradiol in rat models and represent potential G-coupled protein receptors which could lead to the induction of endocrine-signaling independent DMCs.

The online version of this article includes the following figure supplement(s) for figure 6:

**Figure supplement 1.** Differential expression of potential endocrine-signaling independent DMCs.

pathways, including 124 of 800 genes that regulate protein degradation and 5 of 9 genes involving 1-phosphatidylinositol-3 kinase activities, linked to the PI3K/AKT signaling pathway (*Figure 6a*).

Previous reports have established a link between ubiquitin-like protease pathways and classical ER signaling (*Beamish and Frick, 2022*; *Kabir et al., 2015*), suggesting that genes related to ubiquitin-like protease pathways that have promoters lacking EREs may be regulated by factors that are themselves encoded by genes with promoters containing EREs, and so could be indirectly activated or repressed by disruption of classical ER signaling. However, involvement of PI3K/AKT pathway signaling has previously been linked to estrogen signaling through non-canonical G-protein coupled receptors such as GPER1, which was sufficient to induce estrogen signaling in ER KO mouse cell lines which lack the capacity for classical ER signaling (*Filardo et al., 2002*; *Molina et al., 2017*). To determine if our data support the suggestion that the non-canonical BPS-induced changes were linked to involvement of PI3K/AKT pathway genes, we performed GO analysis on the 1890 DEGs identified in PGCLCs

in which epimutations appeared to be induced independent of classical ER signaling. Interestingly, we detected no apparent involvement of pathways involving ubiquitin-like proteases, but we did detect differential expression of four of seven genes associated with a pathway involving phosphatidylinositol-4 phosphatase signaling which intersects with the PI3K/AKT signaling pathway (*Figure 6b*). Finally, while we did not detect expression of *Gper1* transcripts, we did observe differential expression of genes encoding other less well studied G-protein-coupled receptors, including *Gprc5a*, *Gprc5b*, *Gpr89*, *Gpr107*, and *Gpr161* in all four cell types exposed to BPS. These receptors have all been previously shown to bind either BPA or 17β-estradiol in rat models and could be potential targets of non-canonical BPS-induced estrogen signaling (*Figure 6c*) as described in the following links: (*Rat Genome Database, 2024a*; *Rat Genome Database, 2024b*; *Rat Genome Database, 2024c*; *Rat Genome Database, 2024d*; *Rat Genome Database, 2024e*).

## Propagation of BPS-induced epimutations during transitions in cell fate

Transitions between pluripotent and germ cell fates, or vice versa, are accompanied by large-scale epigenetic reprogramming in vivo (*Cantone and Fisher, 2013*; *Lee et al., 2014*; *Santos et al., 2002*; *Sanz et al., 2010*), and these are recapitulated during similar transitions induced in vitro (*Ishikura et al., 2016*). To determine the extent to which BPS-induced epimutations persist during a pluripotent to germline transition in vitro, we first exposed iPSCs to 1 μM BPS and then differentiated the exposed iPSCs first into epiblast-like cells (EpiLCs) and then into PGCLCs, recapitulating the early germline epigenetic reprogramming event that normally occurs in vivo (*Kurimoto and Saitou, 2018*; *Ohta et al., 2017*; *Figure 7a*). We then used genome-wide analyses by EM-seq and RNA-seq to compare numbers of exposure-specific DMCs and DEGs, respectively, in PGCLCs derived from BPS-exposed iPSCs with those in the directly exposed iPSCs and found lower, but still substantial numbers of both (28,168 vs 38,105 DMCs, and 1437 vs 1637 DEGs in the derived PGCLCs compared to the directly exposed iPSCs, respectively; *Figure 7b and c*).

When we compared the specific DMCs and DEGs that were detected in the BPS-exposed iPSCs with those that were detected in the subsequently derived PGCLCs, we found that >90% of each were not conserved during the pluripotent to germline transition in cell fate. Specifically, only 3.7% of the DMCs and 8.4% of the DEGs detected in the BPS-exposed iPSCs were also detected in the PGCLCs derived from the exposed iPSCs. Among the small portion of specific 138 DEGs (*Supplementary file 1*) that were conserved from exposed iPSCs to derived PGCLCs, several (12.32%) (*Cdkn1a*, *Ccnd2*, *Plk2*, *Tgfbr1*, *Gadd45g*, *Lck*, *Ltbr*, *Mad2l1*, *Ap3m2*, *Ctsz*, *Tcirg1*, *Gusb*, *Id2*, *Lefty2*, *Gstm7*, *Acsl1*, *Slc39a14*) were involved in cell cycle and apoptosis pathways which could potentially be linked to cancer development (*Figure 7—figure supplement 1*). Thus, of the 38,105 DMCs and 1637 DEGs induced by exposure of iPSCs to 1 μM BPS, 36,688 and 1499, respectively, did not persist during differentiation of iPSCs to form PGCLCs, and so were apparently corrected by germline reprogramming. Simultaneously however, 26,752 novel DMCs and 1299 novel DEGs appeared in the derived PGCLCs that were not present in the BPS-exposed iPSCs, so were apparently generated de novo during the germline reprogramming process. These results are consistent with the notion that germline epigenetic reprogramming corrected many of the epimutations that were present in the BPS-exposed pluripotent cells from which they were derived, but that exposure of cells to EDCs may disrupt the underlying chromatin landscape in a way that then interferes with subsequent reprogramming such that in addition to correcting many previously induced epimutations, the germline reprogramming process, acting on a disrupted chromatin landscape, also generates many novel epimutations de novo during the pluripotent to germline transition.

## Discussion

While nearly 20 years of research on the effects of exposure of live animals to various EDCs and other environmental disruptive influences has clearly established the potential to perturb the epigenome in ways that can dysregulate normal gene expression patterns and predispose the development of disease states, the molecular mechanisms underlying these phenomena have remained largely undefined. Thus, the manner in which environmental exposures introduce biochemical changes in epigenetic programming, how such disruptions can be propagated within a tissue or throughout the soma of an exposed individual or enter that individual's germ line, or how these disruptions predispose

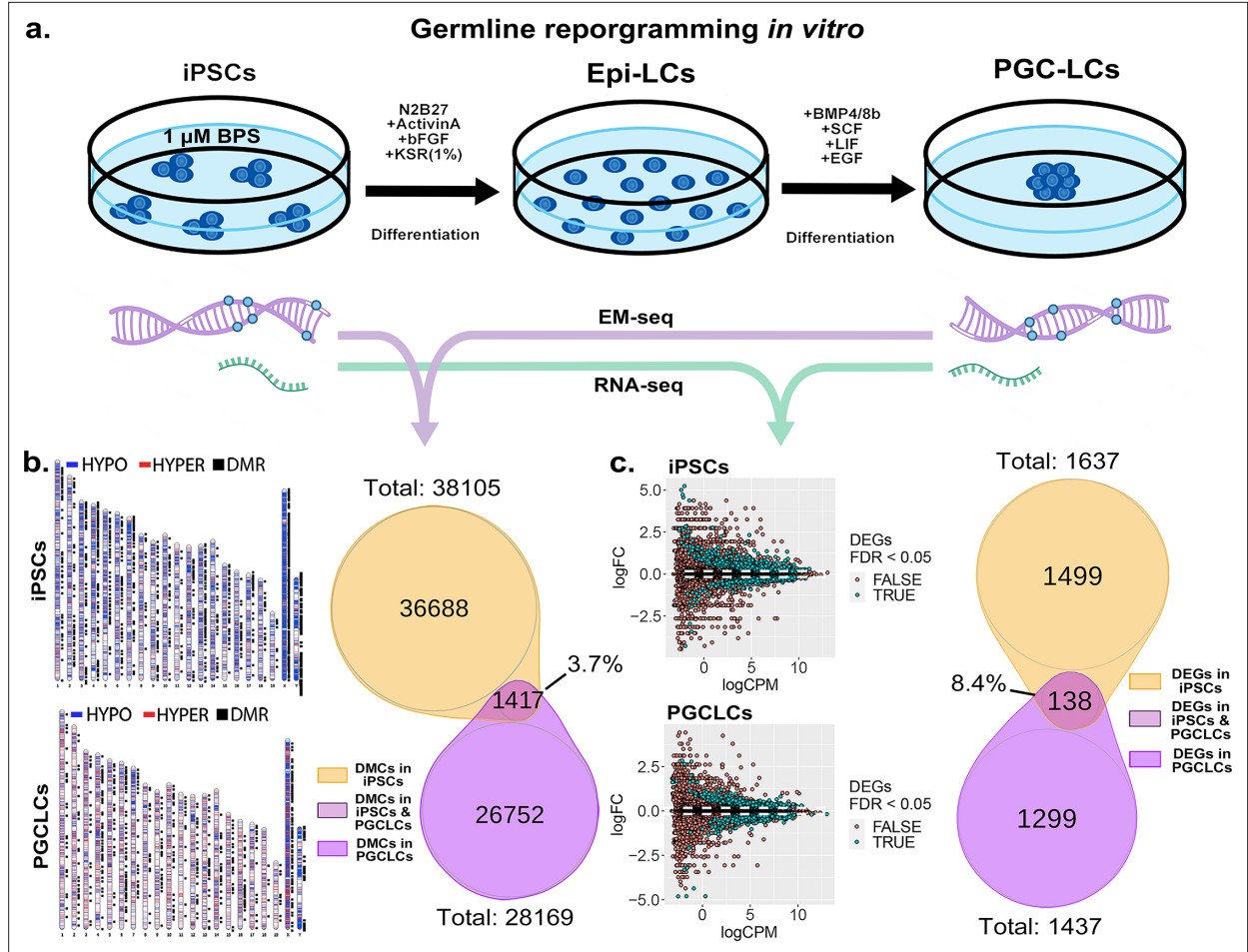

**Figure 7.** Persistence of BPS-induced epimutations through recapitulation of early germline reprogramming in vitro. (**a**) Schematic illustrating derivation of PGCLCs from iPSCs in vitro. iPSCs are first induced to form EpiLCs which are then induced to form PGCLCs. iPSCs were exposed to either ethanol +1 µM BPS or ethanol (carrier) only, then induced to undergo transitions to form EpiLCs and then PGCLCs. (**b**) DNA samples from BPS-exposed or control iPSCs as well as subsequently derived PGCLCs were assessed for exposure-specific DNA methylation epimutations by EM-seq. BPS-treated iPSCs showed 38,105 DMCs and subsequently derived PGCLCs showed 28,169 DMCs. Of those, only 1417 (3.7%) of the DMCs were conserved from the BPS-exposed iPSCs to the subsequently derived PGCLCs. (**c**) RNA samples from BPS-exposed or control iPSCs and subsequently derived PGCLCs were assessed for global gene expression patterns by RNA-seq. BPS-treated iPSCs showed 1637 exposure-specific DEGs and subsequently derived PGCLCs showed 1437 exposure-specific DEGs. Of those, only 138 (8.4%) were conserved from the BPS-exposed iPSCs to the subsequently derived PGCLCs.

The online version of this article includes the following source data and figure supplement(s) for figure 7:

**Source data 1.** Quality control metrics for EM-seq data.

**Figure supplement 1.** KEGG pathway analysis of DEGs detected in both iPSCs exposed to BPS and PGCLCs derived from the exposed iPSCs.

**Figure supplement 2.** Consistency among iPSC and ancestrally exposed PGC-LC replicates and variation between RNA-seq and EM-seq experimental and control groups.

**Figure supplement 3.** Relative expression of markers for PGCLC induction from iPSCs.

**Figure supplement 4.** Quality control for RNA-seq data.

disease states in that individual, or how a resulting prevalence of epimutations can be transmitted to multiple subsequent generations despite generational epigenetic reprogramming, are all mechanistic questions that have remained unanswered despite more than 5000 publications on this topic since the early 2000s. The vast majority of those publications have described experiments in animal models – typically rodents. While informative, such live animal studies require months to years to complete, are cost-, labor-, and animal-intensive, and have not yielded substantial insight into the molecular mechanisms underlying the deleterious effects of environmentally induced epimutations.

In vitro model systems have proven to be very useful tools for deciphering molecular mechanisms related to normal development and homeostasis, or disruptions of those processes predisposing onset of disease. Examples include cell culture systems used to study developmental processes such as X-chromosome inactivation in mammals (*Almeida et al., 2017*; *Patrat et al., 2009*) or early embryogenesis (*Bao et al., 2022*; *Lau et al., 2022*), or to elucidate the cellular and molecular etiology of many different diseases based on the now popular 'disease-in-a-dish' approach (*Davaapil et al., 2020*; *Song et al., 2023*). While in vitro cell culture systems are, by nature, devoid of a majority of the physiological complexities present within the intact organism in vivo, this provides an actual advantage in that it facilitates deconvolution of those complexities while unequivocally focusing on only certain, specific aspects or cell types. However, this ultimately warrants validation of results discerned from studies of in vitro models to ensure they also reflect functions ongoing in the more complex and heterogeneous environment of the intact animal in vivo.

We assembled an in vitro system in which we combined the three cell types normally involved in (i) the initial exposure to disruptive environmental effects (somatic cells), (ii) the first phase of generational epigenetic reprogramming in the preimplantation embryo (pluripotent cells), and (iii) the second phase of generational reprogramming in the developing germ line (germ cells). This system allowed us to gain insights into multiple mechanistic aspects of environmentally induced epimutagenesis including the initial induction of epimutations by exposure of cells to the EDC, BPS, cell-type specific differences in susceptibility to BPS-induced epimutagenesis, factors that appear to contribute to that differential susceptibility, and the mechanism by which an epimutated state may persist across multiple generations despite generational reprogramming.

Here we have shown that exposure of homogeneous populations of cells maintained in culture to an EDC such as BPS can induce epimutagenesis and dysregulation of gene expression, and that this occurs in a partially additive dose-dependent manner, although the quantitative effect appears to plateau at higher doses. This afforded us the opportunity to test the relative effects of direct exposure of different specific cell types to a similar dose of BPS with no confounding effects imposed by interactions with other cell types or systemic effects of the sort that normally occur in the in vivo context. We found that somatic, pluripotent, and germ cell types do indeed show differences in direct susceptibility to induction of epimutations following exposure to BPS, both in terms of the quantity of epimutations induced and qualities of the resulting epimutations such as a predominance of hyper- versus hypomethylation, and the genomic locations where such epimutations tend to occur. Our results suggest that these differences reflect distinctions in the normal status of each cell type at the time of exposure. Thus, somatic, pluripotent, and germ cell types differ in the normal status of genome-wide patterns of accessible versus inaccessible chromatin, global DNA methylation, and expression of relevant endocrine receptors. Interestingly, our data suggest that all of these parameters can influence cell-type specific susceptibility to induction of epimutations by exposure to BPS, and that different combinations of these variables ultimately determine the quantity and quality of epimutations induced by exposure to BPS. We note that while our exposure dose of 1 μM of BPS was below that deemed safe by the EPA for exposure of humans to the similar EDC, BPA, that same dose may exert greater effects when used to expose cells in culture in the absence of any sort of mitigating metabolic effects that may accrue in intact animals or humans. Indeed, the potential to quantify the epimutagenic effects of different doses of an EDC on different cell types as shown in our study could be used in future studies to assess the relative effects of a specific dose of an EDC on a specific cell type when that cell type is exposed either in a homogeneous culture or within an intact animal.

Endogenous fetal germline cells (e.g. PGCs and prospermatogonia in males) normally display lower levels of global DNA methylation and higher levels of chromatin accessibility than pluripotent or somatic cell types (*Hajkova, 2011*), yet PGCLCs developed fewer epimutations following exposure to BPS than the other cell types. This appears to be explained by the fact that germline cells do not express either estrogen receptor, whereas pluripotent and somatic cell types express one or both of the two estrogen receptors, apparently rendering them more susceptible to induction of epimutations following exposure to the estrogen mimetic, BPS. This is consistent with the notion that EDCs impose disruptive effects by interfering with canonical endocrine signaling pathways, and that notion is further supported by our observation that, in the cell types that express one or both estrogen receptors, we observed a prevalence of BPS-induced epimutations in genomic regions containing nearby

EREs, whereas in PGCLCs, which do not express either estrogen receptor, we did not observe a strong correlation between the locations of BPS-induced epimutations and genomic EREs.

Nevertheless, we did observe induction of epimutations in PGCLCs exposed to BPS, suggesting that while cell types expressing relevant endocrine receptors may display elevated susceptibility to EDC-induced epimutagenesis presumably based on disruption of canonical endocrine signaling, this is not the only mechanism by which EDCs can induce epimutations. Indeed, our results support the suggestion that there may be at least two types of epimutations induced by exposure to EDCs – one that is independent of canonical endocrine signaling and so occurs in all cell types regardless of expression of relevant endocrine receptors or the genomic location of relevant HREs, and another that does involve disruption of canonical endocrine signaling and so is elevated in cell types expressing relevant endocrine receptors and predisposes the induction of epimutations in genomic regions near relevant HREs. Beyond this, a comparison of the quantities of BPS-induced epimutations in pluripotent and somatic cell types, which both express one or both estrogen receptors, but which differ with respect to elevated genome-wide chromatin accessibility in pluripotent cells compared to much more limited chromatin accessibility in somatic cell types (*Cantone and Fisher, 2013*; *Santos et al., 2002*), suggests that chromatin accessibility also contributes to susceptibility to EDC-induced epimutagenesis.

One of our most intriguing findings was that, in addition to estrogen-receptor expressing somatic and pluripotent cell types developing more epimutations than non-estrogen-receptor expressing germ cells in response to exposure to BPS, the occurrence of epimutations in the former (somatic and pluripotent) cell types was predominantly in enhancer regions whereas that in the latter (germ) cell type was predominantly in gene promoters. As noted above, there was a significant increase in the occurrence of BPS-induced epimutations near EREs in somatic and pluripotent cell types compared to germ cells. A genome-wide analysis confirmed that ERE half-sites occur more frequently in enhancer regions than in promoter regions (*Table 3* and *Figure 3—figure supplement 4*) supporting our suggestion that the higher prevalence of BPS-induced epimutations we observed in enhancer regions in somatic and pluripotent cell types was due to disruption of canonical hormone signaling.

Our finding that PGCLCs, which do not express endocrine receptors, still accrued epimutations following exposure to BPS indicates that in addition to inducing epimutations by disrupting canonical endocrine signaling pathways, exposure to EDCs can induce epimutations via mechanisms that do not involve disruption of such pathways. This notion is further supported by the occurrence of BPS-induced epimutations in genomic regions that do not include nearby EREs, even though a prevalence of the epimutations induced in PGCLCs were located in gene promoters. Pathway analysis of the genes in which promoter-region epimutations were induced by exposure of PGCLCs to BPS suggests an apparent endocrine-signaling independent group of genes that can become disrupted by exposure to an EDC via non-canonical signaling which may occur through the PI3K/AKT pathway via G-protein-coupled receptors. While we observed expression of a number of G-protein-coupled receptors known to bind to either 17β-estradiol or BPA, further studies will be required to determine if any of these contribute to susceptibility to BPS epimutagenesis.

As expected, disruption of normal epigenetic programming induced by exposure of cells to BPS also led to dysregulation of gene expression. Thus, RNA-seq analysis detected differentially expressed genes in all cell types exposed to BPS. Surprisingly, the greatest number of DEGs was detected in PGCLCs which was the cell type that developed the lowest overall number of epimutations in response to exposure to BPS. However, PGCLCs showed a higher prevalence of BPS-induced epimutations in gene promoter regions than was observed in any of the other cell types, so it appears that promoter-region epimutations impose the most direct impact on gene expression. As noted above, fetal germ cells normally display the lowest level of global DNA methylation of any cell type at any developmental stage. This is characteristic of the epigenetic ground state that accrues uniquely in developing germ cells (*Hajkova, 2011*), and is believed to reflect a generally more accessible chromatin state genome-wide in germ cells than in other cell types. Thus, it may be that the epigenetic ground state in developing germ cells renders this cell type uniquely susceptible to epimutagenesis in gene promoter regions, thereby predisposing a high level of dysregulation of gene expression in response to exposure to an EDC.

Finally, our in vitro system allowed us to follow the fate of BPS-induced epimutations through a major transition in cell fate from pluripotent to germline cells during which a portion of the normal

generational germline epigenetic reprogramming process is known to occur (*Kurimoto and Saitou, 2018*). Thus, when we induced iPSCs previously exposed to 1 µM BPS to differentiate into PGCLCs, we found that a substantial prevalence of epimutations persisted during this transition. However, very few of the specific DMCs or the associated DEGs detected in the BPS-exposed iPSCs persisted in the subsequently derived PGCLCs. Specifically, only 3.7% of DMCs and 8.4% of DEGs were conserved between the BPS-exposed iPSCs and the subsequently derived PGCLCs. This suggests that exposure of cells to EDCs has the potential to disrupt epigenetic programming in a way that then interferes with subsequent generational reprogramming that normally accompanies either germline to pluripotent or pluripotent to germline transitions in cell fate. Recent in vivo studies have suggested that exposure of gestating female mice to the EDC tributyltin results in disruption not simply of the pattern of epigenetic modifications but also of the underlying chromatin landscape (*Chamorro-García et al., 2021*; *Chang et al., 2022*). In this context, our results are consistent with the suggestion that exposure of cells to EDCs can disrupt the underlying chromatin landscape (e.g. the pattern of A and B chromatin compartments) such that when the 'normal' reprogramming machinery then acts on this disrupted landscape it corrects many of the originally induced epimutations, but simultaneously induces many novel epimutations de novo. This would predispose ongoing abnormalities in the underlying chromatin landscape that would, in turn, lead to the recurring correction of many existing epimutations in concert with the genesis of many novel epimutations during each subsequent generation.

We note that, with the exception of our analysis of granulosa cells, all of our studies were carried out in 'male' XY-bearing cells. It remains possible that XX-bearing cells might differ from XY-bearing cells in the way(s) in which they respond to exposure to an EDC. However, the similarity we observed between responses of XX granulosa cells and XY Sertoli cells suggest this may not be the case – at least in cell types in which dosage compensation has been established by X-chromosome inactivation.

Taken together, our results demonstrate the utility of an in vitro cell culture approach for pursuing molecular mechanisms underlying environmentally induced disruption of normal epigenetic programming manifest as the formation of epimutations and dysregulated gene expression. This approach clearly affords unique potential to reveal mechanisms responsible for the initial induction of epimutations in response to direct exposure of cells to disruptive effects such as EDCs, as well as offering potential means to elucidate mechanisms by which environmentally induced epimutations are then propagated within the exposed individual and then to that individual's descendants. Knowledge of these mechanisms will afford the best opportunity to understand how these defects may, in the future, be better diagnosed, treated, and/or prevented. With the ever-expanding catalog of potentially hazardous man-made compounds permeating our environment, it is increasingly important that we recognize the potential dangers such compounds may impose and maximize our ability to protect ourselves from those dangers.

## Materials and methods
### Animal procedures
Mice were used as the primary source of cells for the establishment of cultures for the analysis of differential cell-type specific susceptibility to exposure to the EDC BPS at the cellular level. All mice were euthanized prior to dissection and isolation of the tissue or cell type to be used for experiments. All animals were bred on-site and maintained in one of the UTSA on-campus vivaria under controlled temperature and humidity conditions in a fixed 12 hr light, 12 hr dark cycle with free access to 5V5R extruded food and non-autoclaved conventional RO water. 5V5R has been verified to contain a targeted level of 50 PPM total phytoestrogenic isoflavones (genistein, daidzein, and glycitein) and has been certified for use with estrogen-sensitive protocols limiting the effect of these plant-derived compounds. Euthanasia was performed by trained personnel using continuous $CO_2$ exposure at a rate of 3 L/min until one minute after breathing had stopped. Euthanasia was confirmed by cervical dislocation. Tissue from euthanized mice was used to obtain somatic (Sertoli, granulosa, and mouse embryonic fibroblast [MEF]) cells, and primordial germ cells [PGCs]. Both Sertoli and granulosa cells were isolated from respective male and female mice euthanized at postnatal day 20 (P20). MEFs and PGCs were isolated from fetuses at embryonic day 13.5 postcoitum (E13.5).

# In vitro generation and/or culture of pluripotent, somatic, and germ cells

## Pluripotent cell culture

Male mouse iPSCs were derived from a transgenic mouse line carrying the *tetO-4F2A* cassette obtained from The Jackson Laboratories (011011) (The Jackson Laboratory, ME USA). iPSCs can be induced from essentially any cell type carrying this reprogrammable cassette by exposure to Doxycycline for one week as previously described (*Carey et al., 2010*; *Hochedlinger et al., 2005*). For this project, iPSCs were reprogrammed from MEFs isolated from a single male (XY) E13.5 mouse fetus. Reprogrammed colonies were picked and expanded via sub-passaging. Validation of reprogrammed pluripotency was determined by positive immunocytochemical (ICC) staining for the pluripotent markers POU5F1, SOX2, NANOG, and FUT4 (*Figure 1—figure supplement 5* and *Figure 2—figure supplement 2*). Karyotyping was done on each candidate iPSC line produced to confirm normal chromosome number and XY sex chromosome constitution of each line prior to aliquots being prepared for long-term storage in liquid nitrogen (*Supplementary file 2*). We selected a male (XY) iPSC line to be used for this project. Upon thawing, iPSCs were initially maintained on CF-1 feeder cells for a minimum of three passages. Cells were cultured in DMEM supplemented with 15% fetal bovine serum (FBS) and 1000 U/mL leukocyte inhibitory factor (LIF). Once pluripotent cells were stabilized in culture, they were transitioned to feeder-free conditions and were cultured in N2B27 media supplemented with 2i and LIF for two-three passages prior to use in chemical exposure experiments. A complete list of the media components along with catalog numbers can be found in *Supplementary file 3*.

## Sertoli cell culture

Primary cultures of Sertoli cells were established as previously described (*Karl and Griswold, 1990*). Briefly, whole testes were dissected from ~5 juvenile littermate mice at P20. Following removal of the tunica albuginea from each individual testis, the bundles of seminiferous tubules were physically chopped using a sterile razor blade to break down the coiled structure of tubules increasing the surface area and rendering them easier to digest. These shortened fragments of seminiferous tubules were then digested in a mixture of 2.5% trypsin and 6.64 mg/ml DNaseI in DPBS for 25 min at 37 °C. After these enzymes were inactivated, the tubule sections were washed multiple times, then treated with a final enzymatic mixture of collagenase IV (0.70 mg/mL) and DNaseI (6.64 mg/mL) for 10 min to permeabilize the thick collagen layer on the exterior of the seminiferous tubules to allow the Sertoli cells to migrate out away from the tubules in culture. After checking the digested tubules under a microscope to confirm the collagen layer had been permeabilized, the digested tubules were spun down to wash away the enzymes, and then resuspended in Sertoli cell media containing retinoic acid from ScienCell (4521) (ScienCell Research Laboratories, CA USA). The digested tubules were plated into six culture flasks in order to have three replicates of both control and treated cells for each exposure experiment. To remove contaminating germ cells from this primary culture, the cells were treated with hypotonic shock treatment on the morning of the second day of culture, and the enriched Sertoli cells were washed and allowed to recover with fresh media for at least two hours. The enriched primary cultures of Sertoli cells were then ready to be used for chemical exposure experiments. The estimated purity of the culture was >80% based on ICC staining for the Sertoli cell markers SOX9 and WT1 (*Figure 2—figure supplement 2*). A full procedure for the establishment of primary cultures of Sertoli cells can be found in *Supplementary file 4*.

## Granulosa cell culture

Primary cultures of granulosa cells were established by selecting early preantral primary follicles from enzymatically digested ovary tissue as described previously (*Monti and Redi, 2016*; *Roy and Greenwald, 1996*), followed by breaking selected primary follicles down into a single cell suspension that could be plated and maintained. To isolate preantral primary follicles we dissected ovaries from ~5 female littermate mice at P20 and then enzymatically digested the ovary using collagenase IV (560 U/ pair of ovaries) for 25 min at 37 °C with constant agitation. This digestion was stopped by addition of buffer with 0.5% BSA and the mixture was spun down at 60 x *g* for 5 min to pellet the cells. The cells were resuspended in PBS with 0.5% BSA and transferred to a sterile petri dish under a stereomicroscope. Primary follicles were individually picked from the solution using a glass needle with suction

control and moved into a clean droplet of PBS containing 0.5% BSA. Selected primary follicles were then spun down at 1000 x *g* for 5 min. After spinning, the supernatant was removed, and the cells were finger-vortexed to resuspend the pellet. The follicles were then digested to single cell suspension by addition of pre-warmed (37 °C) 0.25% trypsin and incubated for 5 min with regular pipetting followed by pelleting again at 1000 x *g* for 5 min. The trypsin-containing supernatant was removed, and the granulosa cells were resuspended in DMEM supplemented with 15% FBS and plated into ix culture flasks in order to have three replicates of both control and treated cells for each exposure experiment. The oocytes were non-adherent and were washed away on day 2 when changing the media. The estimated purity of the culture was >90% based on ICC staining for the granulosa cell marker FSHR and INHA (*Figure 2—figure supplement 2*). The granulosa cells were then ready to be used for EDC exposure experiments. A full procedure for the establishment of primary cultures of granulosa cells can be found in *Supplementary file 5*.

## Primordial germ cell like cell culture

Mouse iPSCs were differentiated into PGCLCs as previously described (*Hayashi et al., 2011*). Briefly, iPSCs maintained in N2B27 supplemented with 2i and LIF under feeder-free conditions were differentiated to EpiLCs for two days by the addition of activin A and basic fibroblast growth factor (bFGF) to the N2B27 media. After 2 days, the EpiLC intermediate cells were passaged to low adherence round bottom plates for 4 days in GK15 media containing bone morphogenic protein 4 (BMP4), stem cell factor (SCF), epidermal growth factor (EGF), and LIF to induce the differentiation of a subset of cells (2–3%) in the resulting aggregate to become PGCLCs. 6 batches each being made up of six plates were required in order to obtain sufficient cell numbers for three replicates of both control and treated conditions. During this 4-day period, cell aggregates were ready for EDC exposure (see below). Following EDC exposure, cell aggregates were removed from the low adherence round bottom plates within each batch, dissociated into a single cell suspension, and PGCLCs were fluorescence-activated cell sorted (FACS) on a BD FACSAria II in the UTSA Cell Analysis Core to recover cells that were double-positive for FUT4 and ITGB3. A full list of the media components and catalog numbers for the differentiation of iPSCs to PGCLCs along with our FACS gating for double positive FUT/ITGB3 PGCLCs can be found in *Supplementary file 3* and *Figure 2—figure supplement 3*, respectively. Additional data demonstrating validation of PGCLCs by qRT-PCR and ICC staining are shown in *Figure 7—figure supplement 3*.

## Immunocytochemistry (ICC)

Cells were immunolabeled as previously described (*Rodig, 2022*) to validate the purity of cell primary cultures using known cell specific markers (*Figure 2—figure supplement 2*) to detect the presence or absence of relevant endocrine receptors ERα and ERβ at the protein level. Pluripotent (iPS) and somatic (Sertoli and granulosa) cells were grown on 13 mm Thermanox plastic coverslips (174950) prior to fixation, permeabilization, and immunolabeling (Nalge Nunc International, NY USA). Non-adherent germ (PGC and PGCLC) cells were spun down onto poly-L-lysine coated slides (63410–01) at 400 RPM for one minute using a Thermo Shandon CytoSpin III Cytocentrifuge from Rakin (Electron Microscopy Sciences, PA USA & Rakin Biomedical Corporation, MI USA). Cells were fixed with 4% formaldehyde for 10 min at room temperature (RT) and then washed three times for five minutes each with ICC buffer (PBS containing 0.01% Triton X-100 detergent) to permeabilize the cell and nuclear membranes prior to blocking with 5% goat serum which was added to the ICC buffer and incubated for 1 hr at RT. Following blocking, primary antibodies, in ICC buffer, were added to the slides and left to incubate overnight at 4 °C. The following day, slides were washed three times with ICC buffer, and fluorescent secondary antibodies in ICC buffer were then added to the slides and left to incubate in the dark for 1 hr. After secondary antibody incubation, cellular nuclei were stained with DAPI at a 1:1000 dilution in ICC buffer for 7 min, followed by three final washes with ICC buffer for 5 min. Coverslips then were transferred to microscope slides and mounted with 5–10 μL of VECTASHEILD Antifade Mounting Medium (H-1000) and sealed with clear nail polish before being imaged on a Zeiss AXIO Imager.M1 Fluorescence Microscope (Vector Laboratories Inc, CA, USA and Zeiss Group, Oberkochen DE). Images were processed for contrast and brightness enhancement and for the addition of scale bars using Fiji (RRID:SCR_002285; *Schindelin et al., 2012*). Information about the primary

and secondary antibodies along with catalogue numbers and dilutions used can be found in the Key resources table.

## Quantitative RT-PCR (qRT-PCR)

Total RNA was extracted from 3 replicate preps of cells as described previously (*Rio et al., 2010*) and treated with 1.5 U/µg total RNA RQ1 DNase1 (M6101) to remove contaminating genomic DNA (Promega Corporation, WI USA). Fifty ng of cleaned RNA was retrotranscribed with SuperScript III as recommended by the manufacturer (Invitrogen, MA, USA). Primers were designed using Primer-BLAST (RRID:SCR_003095; *Ye et al., 2012*) from NCBI. A complete list of all primer sequences used in this study can be found in *Supplementary file 6*. Relative expression levels of selected genes were assessed by real-time PCR using the PowerTrack SYBR Green Master Mix according to the manufacturer's instructions (Applied Biosystems, MA USA) then run on a QuantStudio 5 Real-Time PCR System (Thermo Fisher Scientific, MA, USA) and analyzed with QuantStudio Design and Analysis Software. Each sample was normalized based on constitutive expression of the *Gusb* reference gene to obtain the ΔCt [(2-Ctgene-CtGusb)].

## BPS exposure design

Despite numerous studies illustrating the dangers of estrogenic mimetic EDCs, the EPA has currently not published any limits concerning the maximum acceptable dose of BPS below which exposure on a daily basis is considered to be safe. Thus for this study, we established our initial testing range of 1 µM, 50 µM, and 100 µM of BPS based on the limit established for BPA at $\leq 4.44$ µM (*Ribeiro et al., 2019*) selecting 1 dose below the established safe limit and two doses that exceed that limit. For all cell types except PGCLCs, BPS was dissolved in absolute ethanol and added to media gassed with 5% $CO_2$, 5% $O_2$, and balanced $N_2$ and then injected into three replicate sealed T-25 cell culture flasks and left to incubate for 24 hr (*Figure 1—figure supplement 3*). The media was gassed to ensure enrichment to 5% $CO_2$, which is normally regulated by the cell incubator, limiting the potential for the pH to change in the media during this exposure period due to the absorption of $CO_2$ by incubating cells. The concentration of either BPS dissolved in ethanol for treatment groups or ethanol vehicle alone for control groups made up 0.02% of the total media volume. Following 24 hr of exposure, media containing BPS was removed and cells were washed and allowed to recover in fresh untreated media (without BPS or EtOH) for 24 hr prior to harvesting. PGCLCs were not suitable for this exposure paradigm as EpiLCs undergo differentiation to PGCLCs in cell aggregates formed in low-adherence round-bottom 96-well plates and cannot be maintained in T-25 sealed flasks. Therefore, diluted BPS was added to PGCLC media and added to cells in three replicate batches of round-bottom 96-well plates to incubate for 24 hr prior to a shortened wash and 'chase' period of 8 hr prior to cell sorting.

## Methylation beadchip analysis

A total of 1 µg of extracted genomic DNA from each of three replicate exposure experiments for each cell type was bisulfite-converted with the EZ DNA Methylation Kit (D5001) and modified according to the manufacturer's recommendations (Zymo Research, CA USA and Illumina, CA USA). These samples were run on the Infinium Mouse Methylation BeadChip Array following the Illumina Infinium HD Methylation protocol. This beadchip array includes 297,415 cytosine positions within the mouse genome (CpG sites, non-CpG sites, and random SNPs). The methylation score for each CpG is represented as a β-value which is a ratio of the fluorescence intensity ranging between 0 (unmethylated) and 1 (methylated). Arrays were scanned by HiScan (Illumina, CA, USA). Quality control (*Figure 2—figure supplement 4*) and downstream data processing of the.idat files was using the Sensible Step-wise Analysis of DNA Methylation BeadChips (SeSAMe) Bioconductor package (*Ding et al., 2023*; *Triche et al., 2013*; *Zhou et al., 2022*; *Zhou et al., 2018*). DNA methylation levels of differentially methylated cytosines (DMCs) are determined using mixed linear models. This general supervised learning framework identifies CpG loci whose differential methylation is associated with known control vs. treated co-variates. CpG probes on the array were defined as having differential changes that met both p-value and FDR $\leq 0.05$ significant thresholds between treatment and control samples for each cell type analyzed. Additionally, we followed up our DNA methylation analysis of individual dCpGs by identifying differentially methylated regions (DMRs). DMRs were created by grouping all CpGs measured on the array into clusters using Euclidean distance (*Ding et al., 2023*; *Triche et al., 2013*; *Zhou et al., 2022*; *Zhou*

*et al., 2018*). The p-values from the differential methylation of individual CpGs within the resulting CpG clusters were aggregated, and clusters were then filtered selecting for regions that contained a p-value ≤ 0.05 (*Ding et al., 2023*; *Triche et al., 2013*; *Zhou et al., 2022*; *Zhou et al., 2018*).

## RNA-seq

Total RNA was extracted from 3 replicate preps of cells using Trizol as previously described (*Rio et al., 2010*). Contaminating genomic DNA was removed by RQ1 DNase (M6101) treatment according to the manufacturer's instructions (Promega Corporation, WI USA). RNA concentration was determined using Qubit (Q32855) and RNA integrity (RIN) scores were determined using tape station (5067–5576) according to the manufacturer's instructions (Thermo Fisher Scientific Inc, MA, USA and Agilent Technologies, Inc CA, USA). Strand-specific RNA-seq libraries were prepared with the NEBNext Ultra II Directional RNA Library Prep Kit for Illumina sequencing (E7760S) according to the manufacturer's protocol (New England Biolabs, MA, USA). Briefly, this process consisted of poly(A) RNA selection, RNA fragmentation, and double stranded cDNA generation using random oligo(dT) priming followed by end repair to generate blunt ends, adaptor ligation, strand selection, and polymerase chain reaction amplification to generate the final library. Distinct index adaptors were used for multiplexing samples across multiple sequencing lanes. Sequencing was performed on an Illumina NovaSeq 6000 instrument yielding sequences of paired end 2x50 base pair runs. Demultiplexing was performed with the Illumina Bcl2fastq2 program (Illumina, CA, USA).

## RNA-seq expression analysis

The quality of the fastq reads was checked using FastQC (RRID:SCR_014583; *Andrews et al., 2023*; *de Sena Brandine and Smith, 2019*; *Figure 7—figure supplement 4*). Reads were aligned to the mm10 mouse reference genome using the Rsubreads package to produce read counts (RRID:SCR_016945; *Liao et al., 2019*). These were then used for differential gene expression analysis using the edgeR package (RRID:SCR_012802; *Chen et al., 2016*; *McCarthy et al., 2012*; *Robinson et al., 2010*). Briefly, gene counts were normalized using the trimmed mean of M-value normalization (TMM) method before determining counts per million (CPM) values (*Robinson and Oshlack, 2010*). For a gene to be classified as showing differential gene expression between BPS-treated and EtOH vehicle-only control samples, a threshold of both a Benjamini-Hochberg adjusted p-value ≤0.05 and a false discovery rate (FDR)≤0.05 had to be met.

## EM-Seq

Genomic DNA was extracted from three replicates of cells as previously described (*Sambrook and Russell, 2006*). Smaller fragmented DNA (≤10 kb) and contaminating RNA were removed by cleaning the genomic DNA on a genomic DNA clean and concentrator-10 column (D4011), according to the manufacturer's instructions (Zymo Research, CA, USA). DNA concentration was determined using a Qubit (Q32850) and genomic DNA (100 ng) was sheered using a Bioruptor and the size of the sheered DNA was determined using a TapeStation 4200 according to the manufacturer's instructions in the UTSA Genomics Core (Diagenode Inc, NJ, USA and Agilent Technologies, Inc CA, USA). Large size (470–520 bp) EM-seq libraries were prepared with the NEBNext Enzymatic Methyl-seq kit (E7120S) according to the manufacturer's protocol (New England Biolabs, MA, USA). Briefly, this process consisted of A-tailing, adaptor ligation, DNA oxidation by TET2 initiated by the addition of Fe (II), strand denaturization with formamide, deamination by APOBEC3A, polymerase chain reaction amplification, and bead selection to generate the final libraries. Distinct index adaptors were used for multiplexing samples across multiple sequencing lanes. Sequencing was performed on an Illumina NovaSeq 6000 instrument yielding sequences of paired end 2x150 base pair runs. Demultiplexing was performed with the Illumina Bcl2fastq2 program (Illumina, CA, USA).

## EM-Seq analysis of genome-wide DNA methylation patterns

EM-seq data was processed using the comprehensive wg-blimp v10.0.0 software pipeline (*Lehle and McCarrey, 2023*; *Wöste et al., 2020*). In brief, reads were trimmed prior to initiating the wg-blimp pipeline using Trim Galore (RRID:SCR_011847; https://www.bioinformatics.babraham.ac.uk/projects/trim_galore/). Sequenced reads were aligned to the mm10 genome using gemBS (*King et al., 2020*; *Merkel et al., 2019*; *Schilbert et al., 2020*). The BAM files from alignment underwent a series of QC

tests including read quality scoring by FastQC (RRID:SCR_014583; *Andrews et al., 2023*), overall and per-chromosome read coverage calculation, GC content, duplication rate, clipping profiles by Qualimap (RRID:SCR_001209; *Okonechnikov et al., 2016*), and deduplication by the Picard toolkit (RRID:SCR_006525; *Figure 7—source data 1*). Methylation calling was performed by MethylDackel (*Ryan, 2023a*; https://github.com/dpryan79/MethylDackel, copy archived at *Ryan, 2023b*) and statistically significant DMC/DMR calling was performed by the metilene (*Jühling et al., 2016*) and BSmooth (RRID:SCR_005693; *Hansen et al., 2012*) algorithms. Metilene uses a binary segmentation algorithm combined with a two-dimensional statistical test that allows the detection of DMCs/DMRs in large methylation experiments with multiple groups of samples. BSmooth uses a local-likelihood approach to estimate a sample-specific methylation profile, then computes estimates of the mean differences and standard errors for each CpG to form a statistic similar to that used in a t-test. Finally, potential regulatory regions were identified through the use of MethylSeekR (RRID:SCR006513; *Burger et al., 2013*). Results from the pipeline were displayed in the wg-blimp interactive results web browser that was built using the R Shiny local browser hosting framework (RRID:SCR_001626; *Chang, 2023*).

## Acknowledgements

This work was supported by grants from the NIH to JRM (HD98593 and DA054179), and gifts from the Robert J Kleberg, Jr. and Helen C Kleberg Foundation and the Nancy Hurd Smith Foundation. The authors would like to thank: Michael Klein and the Genomics Core Facility and staff at the University of Utah for their assistance with processing the Illumina Infinium Mouse Methylation Beadchip Array samples; Sean Vargas at the UTSA Genomics Core for his expertise and supervision for utilizing equipment to prepare sequencing libraries; Dr. Sandra Cardona at the UTSA Cell Analysis Core for her expertise and supervision performing FACS to isolate cells; Dr. Brian Herman for use of his fluorescent microscope for the collection of images; and Dr. Brian Bringham for biostatistics advice. This work received computational support from UTSA's HPC cluster Arc, operated by Tech Solutions. Sequencing data in this study was generated at the UT Health Genome Sequencing Facility, which is supported by UT Health San Antonio, NIH-NCI P30 CA054174 (Cancer Center at UT Health San Antonio) and NIH Shared Instrument grant S10OD030311 (S10 grant to NovaSeq 6000 System), and CPRIT Core Facility Award (RP220662). Funding National Institute of Health NICHD P50 (HD98593) John R McCarrey. National Institute of Health NIDA (U01DA054179) John R McCarrey. Robert J and Helen C Kleberg Foundation John R McCarrey. Nancy Hurd Smith Foundation John R McCarrey.

## Additional information

### Funding

| Funder | Grant reference number | Author |
|---|---|---|
| Eunice Kennedy Shriver National Institute of Child Health and Human Development | P50 HD98593 | John R McCarrey |
| National Institute on Drug Abuse | U01 DA054179 | John R McCarrey |
| Nancy Hurd Smith Foundation | | John R McCarrey |
| The Robert J. Kleberg, Jr. and Helen C. Kleberg Foundation | | John R McCarrey |

The funders had no role in study design, data collection and interpretation, or the decision to submit the work for publication.

### Author contributions

Jake D Lehle, Conceptualization, Data curation, Formal analysis, Investigation, Methodology, Writing – original draft; Yu-Huey Lin, Data curation, Investigation; Amanda Gomez, Laura Chavez, Investigation;

John R McCarrey, Conceptualization, Formal analysis, Funding acquisition, Investigation, Writing – original draft, Project administration, Writing – review and editing

**Author ORCIDs**
Jake D Lehle ⓘ http://orcid.org/0000-0002-0833-7068
Yu-Huey Lin ⓘ https://orcid.org/0009-0003-1321-0680
John R McCarrey ⓘ https://orcid.org/0000-0002-5784-9318

**Ethics**
This study was performed in strict accordance with the recommendations in the Guide for the Care and Use of Laboratory Animals of the National Institutes of Health. All of the animals were handled according to approved institutional animal care and use committee (IACUC) protocols (MU015, MU016) of the University of Texas at San Antonio. The protocol was approved by the Institutional Animal Care and Use Committee (Approved protocols - MU015, MU016).

Reviewer #1 (Public review): https://doi.org/10.7554/eLife.93975.4.sa1
Reviewer #2 (Public review): https://doi.org/10.7554/eLife.93975.4.sa2
Author response https://doi.org/10.7554/eLife.93975.4.sa3

## Additional files

**Supplementary files**
• MDAR checklist
• Supplementary file 1. Persisting DEGs in PGCLCs derived from iPSCs exposed to 1uM BPS.
• Supplementary file 2. Validation of normal karyotype analysis of MF5-9-1 iPSCs. iPSCs from reprogrammed MEFs were validated for a normal karyotype prior to use in this project.
• Supplementary file 3. iPSC, EpiLC, and PGCLC Culture Media Components.
• Supplementary file 4. Preparation of primary cultures of Sertoli cells from mice.
• Supplementary file 5. Preparation of primary cultures of granulosa cells from mice protocol.
• Supplementary file 6. qRT-PCR primers.

**Data availability**
Methylation beadchip, EM-seq, and RNA-seq data discussed in this publication have been uploaded and can be obtained from the SRA BioProject PRJNA1026145 and GEO accession GSE252723.

The following datasets were generated:

| Author(s) | Year | Dataset title | Dataset URL | Database and Identifier |
|---|---|---|---|---|
| Lehle JD, Lin Y-H, Gomez A, Chavez L, McCarrey JR | 2024 | DNA methylation data | https://www.ncbi.nlm.nih.gov/bioproject/?term=PRJNA1026145 | NCBI BioProject, PRJNA1026145 |
| Lehle JD, Lin Y-H, Gomez A, Chavez L, McCarrey JR | 2024 | Gene expression data | https://www.ncbi.nlm.nih.gov/geo/query/acc.cgi?acc=GSE252723 | NCBI Gene Expression Omnibus, GSE252723 |

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

# Appendix 1

## Appendix 1—key resources table

| Reagent type (species) or resource | Designation | Source or reference | Identifiers | Additional information |
|---|---|---|---|---|
| Genetic reagent (*M. musculus*) | R26[rtTA]; Col1a1[2lox-4F2A] | The Jackson Laboratory | 011011 | |
| Cell line (*M. musculus*) | CF1 Mouse embryonic fibroblasts, MitC-treated | Thermo Fisher Scientific | A34959 | |
| Chemical compound | Dulbecco's Modified Eagle Medium (DMEM) | Thermo Fisher Scientific | 10313021 | High glucose, pyruvate, no glutamine |
| Chemical compound | Fetal bovine serum (FBS) | Thermo Fisher Scientific | 10439024 | Embryonic stem-cell FBS, qualified, USDA-approved regions |
| Chemical compound | Leukemia inhibitory factor (LIF) | Millipore Sigma | ESG1107 | ESGRO Recombinant Mouse LIF Protein |
| Chemical compound | DMEM/F12 | Thermo Fisher Scientific | 21041025 | No phenol red |
| Chemical compound | Insulin | Millipore Sigma | I1882 | From bovine pancreas |
| Chemical compound | Apo-Transferrin | Millipore Sigma | T1147 | From human |
| Chemical compound | Bovine Serum Albumin (BSA) | Thermo Fisher Scientific | 15260037 | Fraction V (7.5% solution) |
| Chemical compound | Progesterone | Millipore Sigma | P8783 | |
| Chemical compound | Putrescine dihydrochloride | Millipore Sigma | P5780 | |
| Chemical compound | Sodium selenite | Millipore Sigma | S5261 | |
| Chemical compound | Neurobasal | Thermo Fisher Scientific | 12348017 | No phenol red |
| Chemical compound | B-27 | Thermo Fisher Scientific | 12587010 | (50 X), minus vitamin A |
| Chemical compound | Penicillin-streptomycin | Thermo Fisher Scientific | 15070063 | (5,000 U/mL) |
| Chemical compound | GlutaMAX | Thermo Fisher Scientific | 35050061 | (100 X) |
| Chemical compound | 2-Mercaptoethanol | Thermo Fisher Scientific | 21985023 | (1000 X) |
| Chemical compound | CHIR99021 | BioVision | 1677–5 | |
| Chemical compound | PD0325901 | Amsbio | 04-0006-02 | 10 mM in DMSO |
| Chemical compound | Recombinant Human/Murine/Rat Activin A | PeproTech | 120–14 | Insect derived |
| Chemical compound | Recombinant Human FGF-Basic (FGF-2/bFGF) | Thermo Fisher Scientific | 13256–029 | |
| Chemical compound | KnockOut Serum | Thermo Fisher Scientific | 10828028 | |
| Chemical compound | Glasgow's MEM (GMEM) | Thermo Fisher Scientific | 11710035 | |
| Chemical compound | Recombinant human bone morphogenetic protein 4 (BMP-4) | R&D Systems | 314 BP-010 | |
| Chemical compound | Recombinant mouse stem cell factor (SCF) | R&D Systems | 455-MC-010 | |
| Chemical compound | Recombinant human epidermal growth factor (EGF), carrier free (CF) | R&D Systems | 2028-EG-200 | |
| Chemical compound | Dulbecco's phosphate-buffered saline (DPBS) | Thermo Fisher Scientific | 14040133 | |
| Chemical compound | Deoxyribonuclease I (DNaseI) | Millipore Sigma | DN25 | From bovine pancreas |
| Chemical compound | Trypsin (2.5%) | Thermo Fisher Scientific | 15090046 | No phenol red |
| Chemical compound | Soybean trypsin inhibitor | Thermo Fisher Scientific | 17075029 | |
| Chemical compound | Collagenase type IV | Worthington | LS004188 | From Clostridium histolyticum |
| Chemical compound | Sertoli Cell Medium | ScienCell Research Laboratories | 4521 | |
| Chemical compound | Ethanol (EtOH) | Fisher | BP28184 | (200 Proof) |

*Appendix 1 Continued on next page*

*Appendix 1 Continued*

| Reagent type (species) or resource | Designation | Source or reference | Identifiers | Additional information |
|---|---|---|---|---|
| Chemical compound | BSA | Millipore Sigma | A9085 | |
| Chemical compound | Heat inactivated (HI) FBS | Thermo Fisher Scientific | 10082147 | |
| Chemical compound | Trypsin-EDTA (0.25%) | Thermo Fisher Scientific | 25200072 | With phenol red |
| Chemical compound | Bisphenol S (BPS) | Millipore Sigma | 43034–100 MG | |
| Chemical compound | Phenol:Chloroform:Isoamyl Alcohol (25:24:1, v/v) | Thermo Fisher Scientific | 15593031 | |
| Chemical compound | TRIzol | Thermo Fisher Scientific | 15596026 | |
| Chemical compound | Isopropanol | Thermo Fisher Scientific | 327272500 | |
| Chemical compound | Proteinase K Solution (20 mg/mL) | Thermo Fisher Scientific | 25530049 | |
| Chemical compound | MaXtract High Density | Quiagen | 129046 | Phase lock gel tubes |
| Chemical compound | Sodium Acetate Solution | Thermo Fisher Scientific | R1181 | 3 M, pH 5.2 |
| Chemical compound | Glycogen (5 mg/ml) | Thermo Fisher Scientific | AM9510 | |
| Chemical compound | NaCl | Thermo Fisher Scientific | J21618.36 | |
| Chemical compound | Tris base | Millipore Sigma | 77-86-1 | |
| Chemical compound | Ethylenediaminetetraacetic acid (EDTA) | Millipore Sigma | E9884-100G | |
| Chemical compound | Sodium dodecyl sulfate (SDS) | Millipore Sigma | 151-21-3 | |
| Chemical compound | Triton X-100 | Thermo Fisher Scientific | 85111 | |
| Chemical compound | RQ1 DNase | Promega | M6101 | |
| Chemical compound | Propidium iodide | BioLegend | 421301 | FCy 5 µL/$10^6$ cells |
| Antibody | ERα | Thermo Fisher Scientific | MA1-310 | Host: mouse monoclonal, ICC 1:100 |
| Antibody | ERβ | GeneTex | GTX70174 | Host: mouse monoclonal, ICC 1:100 |
| Antibody | INHA | Invitrogen | PA5-13681 | Host: rabbit polyclonal, ICC 1:25 |
| Antibody | FSHR | Affinity | AF5477 | Host: rabbit polyclonal, ICC 1:250 |
| Antibody | SOX9 | Abcam | ab185966 | Host: rabbit monoclonal, ICC 1:100 |
| Antibody | GAPDH | Novus | NB300-221 | Host: mouse monoclonal, ICC 1:100 |
| Antibody | WT1 | Novus | NBP2-67587 | Host: rabbit monoclonal, ICC 1:100 |
| Antibody | FUT4 | GeneTex | GTX34467 | Host: rabbit monoclonal, ICC 1:50 |
| Antibody | NANOG | Abcam | ab80892 | Host: rabbit polyclonal, ICC 1:100 |
| Antibody | POU5F1 | Abcam | ab19857 | Host: rabbit polyclonal, ICC 1:200 |
| Antibody | SOX2 | Abcam | ab97959 | Host: rabbit polyclonal, ICC 1:200 |
| Antibody | ID4 | Thermo Fisher Scientific | PA5-26976 | Host: rabbit polyclonal, ICC 1:50 |
| Antibody | AR | Santa Cruz | sc-7305 | Host: mouse monoclonal, ICC 1:50 |
| Antibody | PPARγ | Santa Cruz | sc-7273 | Host: mouse monoclonal, ICC 1:200 |
| Antibody | RXRα | Invitrogen | 433900 | Host: mouse monoclonal, ICC 1:200 |
| Antibody | PRDM1 | Thermo Fisher Scientific | 14-5963-82 | Host: rat monoclonal, ICC 1:50 |
| Antibody | Goat Anti Mouse Alexa 647 | Abcam | ab150119 | Host: goat polyclonal, ICC 1:200 |
| Antibody | Goat Anti Rabbit Alexa 488 | Abcam | ab150081 | Host: goat polyclonal, ICC 1:1000 |
| Antibody | Goat Anti Rabbit Alexa 647 | Abcam | ab150179 | Host: goat polyclonal, ICC 1:200 |
| Antibody | Goat Anti Rat Alexa 647 | Thermo Fisher Scientific | A21247 | Host: goat polyclonal, ICC 1:200 |
| Antibody | FUT4 (IgM, κ), brilliant violet 421 | BD Horizon | 562705 | Host: mouse monoclonal, FCy 5 µL/$10^6$ cells |

*Appendix 1 Continued on next page*

*Appendix 1 Continued*

| Reagent type (species) or resource | Designation | Source or reference | Identifiers | Additional information |
|---|---|---|---|---|
| Antibody | ITGB3 (IgG), PE | BioLegend | 104307 | Host: hamster monoclonal, FCy 1 µL/10⁶ cells |
| Antibody | IgM, κ Isotype control, brilliant violet 421 | BD Horizon | 562704 | Host: mouse monoclonal, FCy 1.25 µL/10⁶ cells |
| Antibody | IgG Isotype control, PE | BioLegend | 400907 | Host: hamster monoclonal, FCy 1 µL/10⁶ cells |
| Commercial assay or kit | RNA Clean & Concentrator-5 | Zymo Research | R1016 | |
| Commercial assay or kit | Genomic DNA Clean & Concentrator-10 | Zymo Research | D4011 | |
| Commercial assay or kit | EZ DNA Methylation Kit | Zymo Research | D5001 | |
| Commercial assay or kit | SuperScript III One-Step RT-PCR System with Platinum Taq DNA Polymerase | Thermo Fisher Scientific | 12574026 | |
| Commercial assay or kit | PowerTrack SYBR Green Master Mix for qPCR | Thermo Fisher Scientific | A46109 | |
| Commercial assay or kit | Infinium Mouse Methylation BeadChip | Illumina | 20041558 | |
| Commercial assay or kit | RNA ScreenTape & Reagents | Agilent | 5067–5576 | |
| Commercial assay or kit | DNA ScreenTape & Reagents | Agilent | 5067–5583 | |
| Commercial assay or kit | Qubit dsDNA (Broad Range) BR Assay Kit | Thermo Fisher Scientific | Q32850 | |
| Commercial assay or kit | Qubit RNA (high sensitivity) HS Assay Kit | Thermo Fisher Scientific | Q32855 | |
| Commercial assay or kit | NEBNext Ultra II Directional RNA Library Prep Kit for Illumina | New England BioLabs | E7765 | |
| Commercial assay or kit | NEBNext Poly(A) mRNA Magnetic Isolation Module | New England BioLabs | E3370 | |
| Software, algorithm | ZEISS ZEN Microscopy Software | https://www.zeiss.com/microscopy/en/products/software/zeiss-zen.html | ZEN 3.7 | RRID:SCR_013672 |
| Software, algorithm | Primer-BLAST | https://www.ncbi.nlm.nih.gov/tools/primer-blast/ | | RRID:SCR_003095 |
| Software, algorithm | QuantSudtio Design & Analysis Software | https://www.thermofisher.com/us/en/home/technical-resources/software-downloads/quantstudio-3-5-real-time-pcr-systems.html | QuantStudio v1.5.1 | |
| Software, algorithm | Fiji | https://fiji.sc/ | Fiji v1.54f | RRID:SCR_002285; *Schindelin et al., 2012* |
| Software, algorithm | Bfastq2 | https://support.illumina.com/downloads/bcl2fastq-conversion-software-v2-20.html | Bcl2fastq2 v2.20 | |
| Software, algorithm | FastQC | https://www.bioinformatics.babraham.ac.uk/projects/fastqc/ | FastQC 0.12.0 | RRID:SCR_014583; *Andrews et al., 2023*; Smith and de Sena Brandine, 2021 |
| Software, algorithm | Wg-blimp | https://github.com/MarWoes/wg-blimp | Wg-blimp v0.10.0 | *Lehle and McCarrey, 2023*; *Wöste et al., 2020* |

*Appendix 1 Continued on next page*

*Appendix 1 Continued*

| Reagent type (species) or resource | Designation | Source or reference | Identifiers | Additional information |
|---|---|---|---|---|
| Software, algorithm | R-Project for Statistical Computing | http://www.r-project.org/ | R 4.2.1 | Packages: SeSAMe *Ding et al., 2023*; *Triche et al., 2013*; *Zhou et al., 2022*, *Zhou et al., 2018*, stringr RRID:SCR_022813; *Wickham and RStudio, 2022*, kintr RRID:SCR_018533; *Xie et al., 2023*, SummarizedExperiment *Morgan et al., 2023*, ggrepel RRID:SCR_017393; *Slowikowski et al., 2023*, pals *Wright, 2023*, wheatmap *Zhou, 2022*, magrittr *Bache et al., 2022*, ggplot2 RRID:SCR_014601; *Wickham et al., 2023a*, dplyr RRID:SCR_016708; *Wickham et al., 2023b*, tidyr RRID:SCR_017102; *Wickham et al., 2023c* ggvenn RRID:SCR_025300; *Yan, 2023*, RColorBrewer RRID:SCR_016697; *Neuwirth, 2022*, RIdeogram *Hao et al., 2020*, AnnotationDbi RRID:SCR_023487; *Pagès et al., 2023*, Mus.musculus *Team, 2015*, BSgenome.Mmusculus.UCSC.mm10 *Team, 2021*, GenomicRanges RRID:SCR_000025; *Lawrence et al., 2013*, universalmotif *Tremblay, 2023*, memes RRID:SCR_001783; *Nystrom, 2023*, plyranges RRID:SCR_021324; *Lee et al., 2019*, rtracklayer RRID:SCR_021325; *Lawrence et al., 2009*, Rsubread RRID:SCR_016945; *Liao et al., 2019*, edgeR RRID:SCR_012802; *Chen et al., 2016*; *McCarthy et al., 2012*; *Robinson et al., 2010* |

