## [Editor Report · eLife assessment]

This **important** study, characterizing the epigenetic and transcriptomic response of a variety of cell types representative of somatic, germline, and pluripotent cells to BPS, reveals the cell type-specific changes in DNA methylation and the relationship with the genome sequence. The findings are **convincing** and provide a basis for future analyses in vivo. This work should be of interest to biomedical researchers who work on epigenetic reprogramming and epigenetic inheritance.

---

## [Referee Report · Reviewer #1 (Public review)]

In this revised manuscript, authors have conducted epigenetic and transcriptomic profiling to understand how environmental chemicals such as BPS can cause epimutations that can propagate to future generations. They used isolated somatic cells from mice (Sertoli, granulosa), pluripotent cells to model preimplantation embryos (iPSCs) and cells to model the germline (PGCLCs). This enabled them to model sequential steps in germline development, and when/how epimutations occur. The major findings were that BPS induced unique epimutations in each cell type, albeit with qualitative and quantitative cell-specific differences; that these epimutations are prevalent in regions associated with estrogen-response elements (EREs); and that epimutations induced in iPSCs are corrected as they differentiate into PGCLCs, concomitant with the emergence of de novo epimutations. This study will be useful in understanding the multigenerational effects of EDCs, and underlying mechanisms.

Strengths include:

(1) Using different cell types representing life stages of epigenetic programming and during which exposures to EDCs have different effects. This progression revealed information both about the correction of epimutations and the emergence of new ones in PGCLCs.

(2) Work conducted by exposing iPSCs to BPS or vehicle, then differentiating to PGCLCs, revealed that novel epimutations emerged.

(3) Relating epimutations to promoter and enhancer regions

During the review process, authors improved the manuscript through better organization, clarifying previous points from reviewers, and providing additional data.

---

## [Referee Report · Reviewer #2 (Public review)]

Summary:

This manuscript uses cell lines representative of germ line cells, somatic cells and pluripotent cells to address the question of how the endocrine disrupting compound BPS affects these various cells with respect to gene expression and DNA methylation. They find a relationship between the presence of estrogen receptor gene expression and the number of DNA methylation and gene expression changes. Notably, PGCLCs do not express estrogen receptors and although they do have fewer changes, changes are nevertheless detected, suggesting a nonconical pathway for BPS-induced perturbations. Additionally, there was a significant increase in the occurrence of BPS-induced epimutations near EREs in somatic and pluripotent cell types compared to germ cells. Epimutations in the somatic and pluripotent cell types were predominantly in enhancer regions whereas that in the germ cell type was predominantly in gene promoters.

Strengths:

The strengths of the paper include the use of various cell types to address sensitivity of the lineages to BPS as well as the observed relationship between the presence of estrogen receptors and changes in gene expression and DNA methylation.

Weaknesses:

The weakness, which has been addressed by the authors, includes the fact that exposures are more complicated in a whole organism than in an isolated cell line.

---

## [Author Response]

The following is the authors’ response to the previous reviews.

Reviewing editor’s list of items remaining to be addressed followed by our responses/actions:

(1) The order and organization of supplemental figures and tables is almost impossible to navigate. Please put them in order.

All the sections from the previous Supplementary files have been divided into individual Supplementary files so that each can be referenced without confusion from the text. All of the references in the body of the text and the author responses have been updated to reflect this change.

(2) The question of sample sizes was partially addressed, with authors stating that cell culture work in iPSCs and PGCLCs was done in replicates of 3. Sertoli and granulosa cells were generated from pooled preps - how many individuals, were they littermates?

Sertoli and granulosa primary cultures were generated from littermates and each prep used 5 animals (males for Sertoli cells and females for granulosa cells). These changes have been added to the body of the text on pages 39 and 40.

(3) Authors need to discuss the limitations of doing work in triplicates. Their PCA (Supplement Figure 9) reveals that in several cases samples from the same treatment were not discriminated by PC1 and/or PC2. This is especially true in e and f, the variance of which was explained by PC1 for cell type, but for which treatments showed poor discrimination by PC2. Some discussion of the limitations of sample size should be provided.

Additional text has been added to what is now Supplementary file 15 to acknowledge this limitation imposed by the limited number of replicates (three) and the ability to resolve the differences in treatments by PCA in subplots e and f. However, we also note that the differences were sufficient to identify significant DMCs/DMRs/DEGs.

Reviwer 2 also noted a potential weakness that “exposures are more complicated in a whole organism than in an isolated cell line.”

We note that in our revised manuscript we included wording noting that despite the advantages of using an in vitro approach to deduce underlying molecular mechanisms, results of such in vitro studies “ultimately warrant validation of results discerned from studies of in vitro models to ensure they also reflect functions ongoing in the more complex and heterogeneous environment of the intact animal in vivo.” Thus we have endeavored to acknowledge the reviewer’s point.

**Reviewer #1 (Public Review):**
Critiques/Comments:(1) A problem with in vitro work is that homogeneous cell lines/cultures are, by nature, absent from the rest of the microenvironment. The authors need to discuss this.

[Addressed on pages: 24-25] – We have added two sentences to the second paragraph of the Discussion section in which we now acknowledge this concern, but also point out that in vitro models of this sort also provide an experimental advantage in that they facilitate a deconvolution of the extensive complexity resident within the intact animal. Nevertheless, we acknowledge that this deconvolution requires ultimate validation of findings obtained within an in vitro model system to ensure they accurately recapitulate functions that occur in the intact animal in vivo.

In response to Reviewer 2’s stated weakness of our study that “The weakness includes the fact that exposures are more complicated in a whole organism than in an isolated cell line,” please note that this added text includes the statement that despite the advantages of using an in vitro approach to deduce underlying molecular mechanisms, results of such in vitro studies “ultimately warrant validation of results discerned from studies of in vitro models to ensure they also reflect functions ongoing in the more complex and heterogeneous environment of the intact animal in vivo.” Thus we have endeavored to acknowledge the reviewer’s point.

(2) What are n's/replicates for each study? Were the same or different samples used to generate the data for RNA sequencing, methylation beadchip analysis, and EM-seq? This clarification is important because if the same cultures were used, this would allow comparisons and correlations within samples.

Addressed on pages: 39-45 and in new Supplementary file 15 – Additional text has been added in the Methods section to indicate that all samples involving cell culture models which include iPSCs and PGCLCs came from a single XY iPS cell line aliquoted into replicates and all primary cultures which included Sertoli and granulosa cells were generated from pooled tissue preps from mice and then aliquoted into replicates. Finally, all experiments in the study were performed on three replicates. Because this experimental design did indeed allow for comparisons among samples, we have added a new Supplementary file 15

which displays PCA plots showing clustering among control and treatment datasets, respectively, as well as distinctions between each cluster representing each experimental condition.

(3) In Figure 1, it is interesting that the 50 uM BPS dose mainly resulted in hypermethylation whereas 100 uM appears to be mainly hypomethylation. (This is based on the subjective appearance of graphs). The authors should discuss and/or present these data more quantitatively. For example, what percentage of changes were hypo/hypermethylation for each treatment? How many DMRs did each dose induce? For the RNA-seq results, again, what were the number of up/down-regulated genes for each dose?

Addressed on pages: 6-7 and in new Supplementary files 1-3 – The experiment shown in Figure 1 was designed to (1) serve as proof of principle that cells maintained in culture could be susceptible to EDC-induced epimutagenesis at all, (2) determine if any response observed would be dose-dependent, and (3) identify a minimally effective dose of BPS to be used for the remaining experiments in this study (which we identified as 1 μM). We agree that it is interesting that the 50 µM dose of BPS induced predominantly hypermethylation changes whereas the 1 µM and 100 µM doses induced predominantly hypomethylation changes, but are not in a position to offer a mechanistic explanation for this outcome at this time. As the results shown satisfied our primary objectives of demonstrating that exposure of cells in culture to BPS could indeed induce DNA methylation epimutations, that this occurs in a dose-dependent manner, and that a dose of as low as 1 µM of BPS was sufficient to induce epimutagenesis, the data obtained satisfied all of the initial objectives of this experiment. That said, in response to the reviewer’s request we have now added text on pages 6-7 alluding to new Supplementary files 1-3 indicating the total number of DMCs and DMRs, as well as the number of DEGs, detected in response to exposure to each dose of BPS shown in Figure 1, as well as stratifying those results to indicate the numbers of hyper- and hypomethylation epimutations and up- and down-regulated DEGs induced in response to each dose of BPS. While, as noted above, investigating the mechanistic basis for the difference in responses induced by the 50 µM versus 1 and 100 µM doses of BPS was beyond the scope of the study presented in this manuscript, we do find this result reminiscent of the “U-shaped” response curves often observed in toxicology studies. Importantly, this result does demonstrate the elevated resolution and specificity of analysis facilitated by our in vitro cell culture model system.

(4) Also in Figure 1, were there DMRs or genes in common across the doses? How did DMRs relate to gene expression results? This would be informative in verifying or refuting expectations that greater methylation is often associated with decreased gene expression.

Addressed on pages: 6-7 and new Supplementary files 1-6 – In general, we observed a coincidence between changes in DNA methylation and changes in gene expression (Supplementary files 1-3). Pertaining directly to the reviewer’s question about the extent to which we observed common DMRs and DEGs across all doses, while we only found 3 overlapping DMRs conserved across all doses tested, we did find an average of 51.25% overlap in DMCs and an average of 80.45% overlap in DEGs across iPSCs exposed to the different doses of BPS shown in Figure 1. In addition, within each dose of BPS tested in iPSCs, we also found that there was an overlap between DMCs and the promoters or gene bodies of many DEGs (Supplementary file 5). Specifically within gene promoters, we observed a correlation between hypermethylated DMCs and decreased gene expression and hypomethylated DMCs and increased gene expression, respectively (Supplementary file 6).

(5) In Figure 2, was there an overlap in the hypo- and/or hyper-methylated DMCs? Please also add more description of the data in 2b to the legend including what the dot sizes/colors mean, etc. Some readers (including me) may not be familiar with this type of data presentation. Some of this comes up in Figure 4, so perhaps allude to this earlier on, or show these data earlier.

Addressed on pages: 8-9 and new Supplementary file 4 – We observed an average of 11.05% overlapping DMCs between different pairs of cell types, we did not observe any DMCs that were shared among all four cell types. Indeed, this limited overlap of DMCs among different cell types exposed to BPS was the primary motivation for the analysis described in Figure 2. Thus, instead of focusing solely on direct overlap between specific DMCs, we instead examined similarities among the different cell types tested in the occurrence of epimutations within different annotated genomic regions. To better describe this, we have now added additional text to page 9. We have also added more detail to the legend for Figure 2 on page 8 to more clearly explain the significance of the dot sizes and colors, explaining that the dot sizes are indicative of the relative number of differentially methylated probes that were detected within each specific annotated genomic region, and that the dot colors are indicative of the calculated enrichment score reflecting the relative abundance of epimutations occurring within a specific annotated genomic region. The relative score is calculated by iterating down the list of DMCs and increasing a running-sum statistic when encountering a DMC within the specific annotated genomic region of interest and decreasing the sum when the epimutation is not in that annotated region. The magnitude of the increment depends upon the relative occurrence of DMCs within a specific annotated genomic region.

(6) iPSCs were derived from male mice MEFs, and subsequently used to differentiate into PGCLCs. The only cell type from an XX female is the granulosa cells. This might be important, and should be mentioned and its potential significance discussed (briefly).

Addressed on page: 29 – We have added a new paragraph just before the final paragraph of the Discussion section in which we acknowledge that most of the cell types analyzed during our study were XY-bearing “male” cells and that the manner in which XX-bearing “female” cells might respond to similar exposures could differ from the responses we observed in XY cells. However, we also noted that our assessment of XX-bearing granulosa cells yielded results very similar to those seen in XY Sertoli cells suggesting that, at least for differentiated somatic cell types, there does not appear to be a significant sex-specific difference in response to exposure to a similar dose of the same EDC. That said, we also acknowledged that in cell types in which dosage compensation based on X-chromosome inactivation is not in place, differences between XY- and XX-bearing cells could accrue.

(7) EREs are only one type of hormone response element. The authors make the point that other mechanisms of BPS action are independent of canonical endocrine signaling. Would authors please briefly speculate on the possibility that other endocrine pathways including those utilizing AREs or other HREs may play a role? In other words, it may not be endocrine signaling independent. The statement that the differences between PGCLCs and other cells are largely due to the absence of ERs is overly simplistic.

Addressed on page: 11 and in a new Supplementary file 8 – Previous reports have indicated that BPS does not have the capacity to bind with the androgen receptor (Pelch *et al*., 2019; Yang *et al*., 2024). However there have been reports indicating that BPS can interact with other endocrine receptors including PPARγ and RXRα, which play a role in lipid accumulation and the potential to be linked to obesity phenotypes (Gao et al., 2020; Sharma et al., 2018). To address the reviewer’s comment we assessed the expression of a panel of hormone receptors including PPARγ, RXRα, and AR in each of the cell types examined in our study and these results are now shown in a new Supplementary file 8. We show that in addition to not expressing either estrogen receptor (ERa or ERb), germ cells also do not express any of the other endocrine receptors we tested including AR, PPARγ, and RXRα. Thus we now note that these results support our suggestion that the induction of epimutations we observed in germ cells in response to exposure to BPS appears to reflect disruption of non-canonical endocrine signaling. We also note that non-canonical endocrine signaling is well established (Brenker et al., 2018; Ozgyin et al., 2015; Song et al., 2011; Thomas and Dong, 2006). Thus we feel the suggestion that the effects of BPS exposure could conceivably reflect either disruption of canonical or non-canonical signaling in any cell type is well justified and that our data suggests that both of these effects appear to have accrued in the cells examined in our study as suggested in the text of our manuscript.

(8) Interpretation of data from the GO analysis is similarly overly simplistic. The pathways identified and discussed (e.g. PI3K/AKT and ubiquitin-like protease pathways) are involved in numerous functions, both endocrine and non-endocrine. Also, are the data shown in Figure 6a from all 4 cell types? I am confused by the heatmap in 6c, which genes were significantly affected by treatment in which cell types?

Addressed on pages: 19-21 – Per the reviewer’s request, we have added text to indicate that Figure 6a is indeed data from all four cell types examined. We have also modified the text to further clarify that Figure 6c displays the expression of other G-coupled protein receptors which are expressed at similar, if not higher, levels than either ER in all cell types examined, and that these have been shown to have the potential to bind to either 17β-estradiol or BPA in rat models. As alluded to by the reviewer, this is indicative of a wide variety of distinct pathways and/or functions that can potentially be impacted by exposure to an EDC such as BPS. Thus, we have attempted to acknowledge the reviewer’s primary point that BPS may interact with a variety of receptors or other factors involved with a wide variety of different pathways and functions. Importantly, this illustrates the strength of our model system in that it can be used to identify potential impacted target pathways that can then be subsequently pursued further as deemed appropriate.

(9) In Figure 7, what were the 138 genes? Any commonalities among them?

Addressed on page: 22 and in a new Supplementary files 13 and 14 – We have now added a new supplemental Excel file (Supplementary file 13) that lists the 138 overlapping conserved DEGs that did not become reprogrammed/corrected during the transition from iPSCs to PGCLCs. In addition, we have added new text on page 22 and a new Supplementary file 14 which displays KEGG analysis of pathways associated with these 138 retained DEGs. We find that these genes are primarily involved with cell cycle and apoptosis pathways which, interestingly, have the potential to be linked to cancer development which is often linked to disruptions in chromatin architecture.

(10) The Introduction is very long. The last paragraph, beginning line 105, is a long summary of results and interpretations that better fit in a Discussion section.

Addressed on page: 6 – We have now significantly reduced the length and scope of the final paragraph of the Introduction per the reviewer’s recommendation.

(11) Provide some details on husbandry: e.g. were they bred on-site? What food was given, and how was water treated? These questions are to get at efforts to minimize exposure to other chemicals.

Addressed on page: 37 – We have added additional text detailing that all mice used in the project were bred onsite, water was non-autoclaved conventional RO water, and our selection of 5V5R extruded feed for mice used in this study which was highly controlled for the presence of isoflavones and has been certified to be used for estrogen-sensitive animal protocols.

**Reviewer #2 (Public Review):**
Summary:This manuscript uses cell lines representative of germ line cells, somatic cells, and pluripotent cells to address the question of how the endocrine-disrupting compound BPS affects these various cells with respect to gene expression and DNA methylation. They find a relationship between the presence of estrogen receptor gene expression and the number of DNA methylation and gene expression changes. Notably, PGCLCs do not express estrogen receptors and although they do have fewer changes, changes are nevertheless detected, suggesting a nonconical pathway for BPS-induced perturbations. Additionally, there was a significant increase in the occurrence of BPS-induced epimutations near EREs in somatic and pluripotent cell types compared to germ cells. Epimutations in the somatic and pluripotent cell types were predominantly in enhancer regions whereas that in the germ cell type was predominantly in gene promoters.Strengths:The strengths of the paper include the use of various cell types to address the sensitivity of the lineages to BPS as well as the observed relationship between the presence of estrogen receptors and changes in gene expression and DNA methylation.Weaknesses:The weaknesses include the lack of reporting of replicates, superficial bioinformatic analysis, and the fact that exposures are more complicated in a whole organism than in an isolated cell line.Recommendations for the authors: please note that you control which revisions to undertake from the public reviews and recommendations for the authors.
**Reviewer #2 (Recommendations For The Authors):**
Overall, this is an intriguing paper but more transparency in the replicates and methods and a more rigorous bioinformatic treatment of the data are required.Specific comments:(1) End of abstract "These results suggest a unique mechanism by which an EDC-induced epimutated state may be propagated transgenerationally following a single exposure to the causative EDC." This is overly speculative for an abstract. There is only epigenetic inheritance following mitosis or differentiation presented in this study. There is no meiosis and therefore no ability to assess multi- or transgenerational inheritance.

Addressed on page: 2 – We have modified the text at the end of the abstract to more precisely reflect our intended conclusions based on our data. In our view, the ability of induced epimutations to transcend meiosis per se is not as relevant to the mechanism of transgenerational inheritance as their ability to transcend major waves of epigenetic reprogramming that normally occur during development of the germ line. In this regard the transition from pluripotent iPSCs to germline PGCLCs has been shown to recapitulate at least the first portion of normal germline reprogramming, and now our data provide novel insight into the fate of induced epimutations during this process. Specifically, we show that a prevelance of epimutations was conserved during the iPSC à germ cell transition but that very few (< 5%) of the specific epimutations present in the the BPS-exposed iPSCs were retained when those cells were induced to form PGCLCs. Rather, we observed apparent correction of a large majority of the initially induced epimutations during this transition, but this was accompanied by the apparent de novo generation of novel epimutations in the PGCLCs. We suggest, based on other recent reports in the literature, that this is a result of the BPS exposure inducing changes in the chromatin architecture in the exposed iPSCs such that when the normal germline reprogramming mechanism is imposed on this disrupted chromatin template there is both correction of many existing epimutations and the genesis of many novel epimutations. This observation has the potential to explain the long-standing question of why the prevalence of epimutations persists across multiple generations despite the occurrence of epigenetic reprogramming during each generation. Nevertheless, as noted above, we have modified the text at the end of the abstract to temper this interpretation given that it is still somewhat speculative at this point.

(2) Doses used in the experiments. One needs to be careful when stating that the dose used is "below FDA's suggested safe environmental level established for BPA" because a different bisphenol is being used here (BPA vs BPS) and the safe level is that which the entire organism experiences. It is likely that cell lines experience a higher effective dose.

Addressed on pages: 3, 5, and 26 – We have now made a point of noting that our reference to an EPA-recommended “safe dose” of BPA was for humans and/or intact animals. Changes to this effect have been made in the second and sixth paragraphs of the Introduction section. In addition, we have added text at the end of the fourth paragraph of the Discussion section acknowledging that, as the reviewer suggests, the same dose of an EDC could exert greater effects on cells in a homogeneous culture than on the same cell type within an intact animal given the potential for mitigating metabolic effects in the latter. However, we also note that the ability we demonstrated to quantify the effects of such exposures on the basis of numbers of epimutations (DMCs or DMRs) induced could potentially be used in future studies to study this question by assessing the effects of a specific dose of a specific EDC on a specific cell type when exposed either within a homogeneous culture or within an intact animal.

(3) Figure 1: In the dose response, what was the overlap in DMCs and DEGs among the 3 doses? Are the responses additive, synergistic, or completely non-overlapping? This is an important point that should be addressed.

Addressed on page: 6-7 and in Supplementary files 1-5 – Please see our response to Reviewer 1 critique #4 above where we address similar concerns. While we do find overlap among different cell types with respect to the DMCs, DMRs, and DEGs displayed in Figure 1, we found the effect to be only partially additive as opposed to synergistic in any apparent manner. The fold increase in DMCs, DMRs, and DEGs resulting from exposure to doses of 1 μM or 50 μM ranged from 2.5x to 4.4x, which was well below the 50x increase that would have been expected from a strictly additive effect, and the effect increased even less, if at all, in response to exposure to doses of 50 μM versus 100 μM BPS. Finally, as now noted in the Discussion section on page 25, our conclusion is that these results display a limited dose-dependent effect that was partially additive but also plateaued at the highest doses tested.

(4) Methods: How many times was each exposure performed on a given cell type? This information should be in the figure legends and methods. In the case of multiple exposures for a given line, do the biological replicates agree?

Addressed on pages: 39-45 and in new Supplementary file 15 – Please see our response to Reviewer 1 critique #2 where we address similar concerns with newly added text and analysis. We now note repeatedly on pages 39-45 that each analysis was conducted on three replicate samples, and we display the similarity among those replicates graphically in a new Supplementary file 15.

(5) DNA methylation analyses. Very little analysis is presented on the BeadChip array other than hypermethylated/hypomethylated and genomic regions of DMCs. What is the range of methylation changes? Does it vary between hypo vs. hyper DMCs? How many array experiments were performed (biological replicates) and what stats were used to determine the DMCs? Are there DMCs in common among the various cell types? As an example, if more meaningful analysis, one can plot the %5mC over a given array for comparisons between control and treated cell types. For more granularity, the %5mC can be presented according to the element type (enhancers vs promoters).

Addressed on pages: 10 and 39-45 and in new Supplementary files 1-5, 15 – Please see our response to Reviewer 1 critique #2 above where we address similar concerns regarding the number of biological replicates used in this study. DMCs on the Infinium array are identified using mixed linear models. This general supervised learning framework identifies CpG loci at which differential methylation is associated with known control vs. treated co-variates. CpG probes on the array were defined as having differential changes that met both p-value and FDR (≤ 0.05) significant thresholds between treatment and control samples for each cell type analyzed. The range of medians across all samples was 0.0278 to 0.0059 for hypermethylated beta values and -0.0179 to -0.0033 for hypomethylated beta values. As noted above, we did observe an overlap in DMCs between cell types. Thus, we observed an average of 11.05% overlapping DMCs between two or more cell types but we did not observe any DMCs shared between all four cell types. We have added additional text on page 9 and new Supplementary files 1-5 to now more clearly describe that this limited similarity in direct overlap of DMCs was the underlying motivation for the analysis described in Figure 2. Finally, the enrichment dot plots shown in Figure 2 provide the information the reviewer requested regarding the %5mC observed at different annotated genomic element types.

(6) The investigators correlate the number of DMCs in a given cell type with the presence of estrogen receptors. Does the correlation extend to the methylation difference (delta beta) at the statistically different probes?

Addressed in a new Supplementary file 7 – We have added a new Supplementary file 7 in which we provide data addressing this question. In brief, we find that the delta betas of probes enriched at enhancer regions and associated with relative proximity to ERE elements in Sertoli cells, granulosa cells, and iPSCs appear very similar to those associated with DMCs not located within these enriched regions. However, when we compared the similarity of the two data sets with goodness of fit tests, we found these relatively small differences were, in fact, statistically significant based on a two-sample Kolmogorov-Smirnov test. These observed significant differences appear to indicate that there is higher variability among the delta betas associated with hypomethylated, but not hypermethylation changes occurring at DMCs associated with enhancers, potentially suggesting a greater tendency for exposure to BPS to induce hypomethylation rather than hypermethylation changes, at least in these specific regions.

(7) Methylation changes relative to EREs are presented in multiple figures. Are other sequences enriched in the DMCs?

Addressed in a new Supplementary file 11. We profiled the genomic sequence within 500 bp of cell type-specific enriched DMCs that were either associated with enhancer regions in Sertoli, granulosa, or iPS cells or transcription factor binding sites in PGCLCs for the identification of higher abundance motif sequences. We then compared any motifs identified with the JASPAR database to potentially find transcription factors that could be binding to these regions. Interestingly we found that the two most common motifs across all cell types were associated with either the chromatin remodeling transcription factor HMG1A or the pluripotency factor KLF4.

(8) Please present a correlation plot between the methylation differences and the adjacent DEGs. Again, the absence of consideration of the absolute changes in methylation and gene expression minimizes the impact of the data.

Addressed on pages 6, 7, and 17 and in a new Supplementary file 6 – We analyzed the relationship between DMCs at DEGs promoter regions and the corresponding change in expression of that DEG. Our data support a relationship between up-regulated genes showing decreased methylation in promoter regions and down-regulated genes showing increased methylation at promoter regions, although there were some exceptions to this relationship.

(9) EM-Seq is mentioned in Figure 7 and in the material and methods. Where is it used in this study?

Addressed on page 22 – We now note in the text on page 22 that EM-seq was used during experiments assessing the propagation of BPS-induced epimutations during the iPSC à EpiLC à PGCLC cell state transitions to gather higher resolution data of changes to DNA methylation differences at the whole-epigenome level.

References

Brenker C, Rehfeld A, Schiffer C, Kierzek M, Kaupp UB, Skakkebæk NE, Strünker T. 2018. Synergistic activation of CatSper Ca2+ channels in human sperm by oviductal ligands and endocrine disrupting chemicals. *Hum Reprod* 33:1915–1923. doi:10.1093/humrep/dey275

Gao P, Wang L, Yang N, Wen J, Zhao M, Su G, Zhang J, Weng D. 2020. Peroxisome proliferator-activated receptor gamma (PPARγ) activation and metabolism disturbance induced by bisphenol A and its replacement analog bisphenol S using in vitro macrophages and in vivo mouse models. *Environ Int* 134. doi:10.1016/J.ENVINT.2019.105328

Ozgyin L, Erdos E, Bojcsuk D, Balint BL. 2015. Nuclear receptors in transgenerational epigenetic inheritance. *Prog Biophys Mol Biol*. doi:10.1016/j.pbiomolbio.2015.02.012

Pelch KE, Li Y, Perera L, Thayer KA, Korach KS. 2019. Characterization of Estrogenic and Androgenic Activities for Bisphenol A-like Chemicals (BPs): In Vitro Estrogen and Androgen Receptors Transcriptional Activation, Gene Regulation, and Binding Profiles. *Toxicol Sci* 172:23–37. doi:10.1093/TOXSCI/KFZ173

Sharma S, Ahmad S, Khan MF, Parvez S, Raisuddin S. 2018. In silico molecular interaction of bisphenol analogues with human nuclear receptors reveals their stronger affinity vs. classical bisphenol A. *Toxicol Mech Methods* 28:660–669. doi:10.1080/15376516.2018.1491663

Song KH, Lee K, Choi H-S. 2011. Endocrine Disrupter Bisphenol A Induces Orphan Nuclear Receptor Nur77 Gene Expression and Steroidogenesis in Mouse Testicular Leydig Cells. *Endocrinology* 143:2208–2215. doi:10.1210/endo.143.6.8847

Thomas P, Dong J. 2006. Binding and activation of the seven-transmembrane estrogen receptor GPR30 by environmental estrogens: A potential novel mechanism of endocrine disruption. *J Steroid Biochem Mol Biol* 102:175–179. doi:10.1016/j.jsbmb.2006.09.017

Yang Z, Wang L, Yang Y, Pang X, Sun Y, Liang Y, Cao H. 2024. Screening of the Antagonistic Activity of Potential Bisphenol A Alternatives toward the Androgen Receptor Using Machine Learning and Molecular Dynamics Simulation. *Environ Sci Technol* 58:2817–2829. doi:10.1021/ACS.EST.3C09779/ASSET/IMAGES/LARGE/ES3C09779_0004.JPEG